# Diurnal, seasonal, and interannual variations in δ($^{18}$O) of atmospheric O₂ and its application to evaluate natural/anthropogenic changes in oxygen, carbon, and water cycles

Shigeyuki Ishidoya[1], Satoshi Sugawara[2], Atsushi Okazaki[3]

[1]National Institute of Advanced Industrial Science and Technology (AIST), Tsukuba 305-8569, Japan

[2]Miyagi University of Education, Sendai 980-0845, Japan

[3]Chiba University, Chiba 263-8522, Japan

*Correspondence to*: Shigeyuki Ishidoya (s-ishidoya@aist.go.jp)

**Abstract.**

Variations in the $\delta(^{18}O)$ of atmospheric O₂, $\delta_{atm}(^{18}O)$, is an indicator of biological and water processes associated with the Dole-Morita effect (DME). The DME and its variations have been observed in ice cores for paleoclimate studies, however, variations in present-day's $\delta_{atm}(^{18}O)$ have never been detected so far. Here, we present diurnal, seasonal, and interannual variations of $\delta_{atm}(^{18}O)$ based on observations at a surface site in central Japan. The average diurnal $\delta_{atm}(^{18}O)$ cycle reached a minimum during the daytime, and its amplitude was larger in summer than in winter. We found that use of $\delta_{atm}(^{18}O)$ enabled separation of variations of atmospheric $\delta(O_2/N_2)$ into contributions from biological activities and fossil fuel combustion. The average seasonal $\delta_{atm}(^{18}O)$ cycle reached at a minimum in summer, and the peak-to-peak amplitude was about 2 per meg. A box model that incorporated biological and water processes reproduced the general characteristics of the observed diurnal and seasonal cycles. A slight but significant secular increase of $\delta_{atm}(^{18}O)$ by $(0.22 \pm 0.14)$ per meg a$^{-1}$ occurred during 2013–2022. Secular changes in $\delta_{atm}(^{18}O)$ were also simulated by using the box model considering long-term changes of terrestrial gross primary production (GPP), photorespiration, and $\delta(^{18}O)$ of leaf water ($\delta_{LW}(^{18}O)$). We calculated changes of $\delta_{LW}(^{18}O)$ using a state-of-the-art, three-dimensional model, MIROC5-iso. The observed secular increase of $\delta_{atm}(^{18}O)$ was reproduced by the box model that incorporated the isotopic effects associated with the DME from Bender et al. (1994), while the simulated $\delta_{atm}(^{18}O)$ showed secular decrease when the model incorporated the isotopic effects from Luz and Barkan (2011). Therefore, long-term observations of $\delta_{atm}(^{18}O)$ and better understanding of the DME are indispensable, for an application of $\delta_{atm}(^{18}O)$ to constrain long-term changes in global GPP and photorespiration.

# 1 Introduction

The $^{18}O/^{16}O$ ratio of atmospheric $O_2$, $\delta_{atm}(^{18}O)$, is about 24 ‰ higher than that of ocean water (per definition 0 ‰ on the Vienna-Standard Mean Ocean Water (V-SMOW)) due to various processes in the global oxygen and water cycle (e.g. Craig, 1961; Barkan and Luz, 2005). The enrichment of $\delta_{atm}(^{18}O)$ is well known as the Dole-Morita effect (DME) (Dole, 1935; Morita, 1935). The DME is determined from the balance between enrichment of $\delta_{atm}(^{18}O)$ due to discrimination against $^{18}O$ during terrestrial/marine respiratory $O_2$ consumption and the terrestrial/marine photosynthetic $O_2$ flux, for which the $\delta(^{18}O)$ is close to that of ocean water. Bender et al. (1994) (hereafter referred to as "B94") reported that the isotopic effects of dark respiration, photorespiration, and the Mehler reaction associated with terrestrial respiration are 18, 21.2, and 15.3 ‰, respectively, and the terrestrial photosynthetic $O_2$ flux is also affected by discrimination against $^{18}O$ during evapotranspiration (4.4 ‰). (See Table 1 of B94 for a summary of the isotopic effects related to the DME.) Luz and Barkan (2011) (hereafter referred to as "L&B11") also reported the isotopic effects of dark respiration, photorespiration, and evapotranspiration to be 15.8, 22, and 6.5 ‰, respectively. The DME is a useful tool for examining Earth system models because it integrates land and ocean biological and climatic components (e.g., Bender et al, 1994; Luz et al., 1999; Angert et al., 2001; Angert et al., 2003; Hoffman et al., 2004; Barkan and Luz, 2005; Severinghaus et al., 2009; Luz and Barkan, 2011). Some paleoclimate studies have focused on the temporal changes of $\delta_{atm}(^{18}O)$. B94 have reported that the DME is on average lower by 0.05 ‰ than that of present air during the past 130,000 years, and the standard deviation of the DME from the average was ±0.2 ‰. They suggested that the DME was nearly unchanged between glacial maxima and interglacial periods, and the variability is small and may be due to variations of the relative rates of primary production on the land and in the ocean. Severinghaus et al. (2009) have reported the $\delta_{atm}(^{18}O)$ in the Siple Dome ice core, Antarctica, and have found that its variations over the past 60 ka are related to Heinrich and Dansgaard-Oeschger events. They have suggested that the DME is primarily governed by the strength of the Asian and North African monsoons and have confirmed that widespread changes of low-latitude terrestrial rainfall accompany abrupt climate changes.

Hoffman et al. (2004) have developed a model of the DME by combining the results of three-dimensional (3D) models of carbon and oxygen cycles with results of atmospheric general circulation models with built-in water isotope diagnostics and have obtained the average DME of 22.4 to 23.3 ‰. However, they did not simulate temporal or spatial variations of $\delta^{18}O_{atm}$ in the present atmosphere, which have not yet been detected. The diurnal cycle of the atmospheric $O_2/N_2$ ratio at forest sites is caused mainly by activities in the terrestrial biosphere, and the peak-to-peak amplitudes are roughly about 100 per meg (1 per meg is 0.001 ‰) (e.g., Ishidoya et al., 2013a; Battle et al., 2019; Faassen et al., 2023). Diurnal variations of $\delta^{18}O_{atm}$ associated with activities in the terrestrial biosphere are therefore expected to be very small. Keeling (1995) has predicted that $\delta_{atm}(^{18}O)$ should be lower in summer than in winter in both hemispheres by about 2 per meg by assuming a 100 per meg seasonal increase of the atmospheric $O_2/N_2$ ratio due to the input of terrestrial and oceanic photosynthetic $O_2$, which has a $\delta(^{18}O)$ that is lower than $\delta_{atm}(^{18}O)$ by 20 ‰. Seibt et al. (2005), who calculated potential effects of human activity on the DME, have estimated

that global changes of the terrestrial biosphere may have led to a decrease of $\delta_{atm}(^{18}O)$ on the order of 70 per meg over the last 150 years (–0.5 per meg a$^{-1}$). They have estimated that 2/3 of the total decrease is due to a decrease of photorespiration globally accompanied by a 100 µmol mol$^{-1}$ increase of the fraction of atmospheric $CO_2$ during those 150 years. Diurnal, seasonal, and secular changes of $\delta_{atm}(^{18}O)$ in the present atmosphere will therefore be a new indicator of activities of the land and oceanic biospheres, although sufficiently precise measurements of $\delta_{atm}(^{18}O)$ to validate the suggestions by Keeling (1995) and Seibt et al. (2005) have never been reported.

In this study, we present diurnal, seasonal, and secular changes of $\delta_{atm}(^{18}O)$ observed at Tsukuba (TKB), Japan (36° N, 140° E). We then compare the observed changes of $\delta_{atm}(^{18}O)$ at TKB with a one-box model that incorporates the biosphere and water processes associated with the DME. To evaluate the secular changes of water processes, we used an isotope-enabled version of the Model for Interdisciplinary Research on Climate (MIROC5-iso) (Okazaki and Yoshimura, 2019) and calculated the $\delta(^{18}O)$ of leaf water, $\delta_{LW}(^{18}O)$. We suggest some applications of $\delta_{atm}(^{18}O)$, (1) separation of diurnal $\delta(O_2/N_2)$ cycle into contributions from biological activities and fossil fuel combustion, (2) constraint of seasonal $\delta_{LW}(^{18}O)$ cycle, and (3) evaluation of recent secular changes in terrestrial gross primary production (GPP) and photorespiration.

## 2 Methods

### 2.1 Continuous atmospheric measurements of $\delta_{atm}(^{18}O)$ and $\delta(O_2/N_2)$

Air was sampled with a diaphragm pump from an air intake located on the roof of a laboratory building of the National Institute of Advanced Industrial Science and Technology (AIST) at TKB. The gas velocity exceeded 5 m s$^{-1}$ (4 mm i.d. and a flow rate of 4 L min$^{-1}$) at the tip of the air intake, which was high enough to prevent thermally diffusive inlet fractionation (Sturm et al., 2006; Blaine et al., 2006). The sample air was introduced into a 1-L, stainless-steel buffer tank after water vapor in the air had been reduced by using an electric cooling unit at 2 °C. The gas was then exhausted from the buffer tank at a flow rate of about 4 L min$^{-1}$. A small portion of this exhausted gas was introduced into a 3.2-mm (1/8 in.) o.d. stainless-steel tube, and any remaining water vapor was removed using a cold trap at −90 °C. Finally, the remaining sample air was vented through an outlet path at a rate of about 10 mL min$^{-1}$, and a minuscule amount of it was transferred to the ion source (or waste line) of a mass spectrometer (Thermo Scientific Delta V) through a thin, insulated, fused-silica capillary. The reference air was always supplied from a high-pressure cylinder at a flow rate of about 4 mL min$^{-1}$, and a minuscule amount of it was introduced into the ion source (or waste line) of the mass spectrometer through another fused-silica capillary. The standard air, which was supplied from a high-pressure cylinder at a flow rate of 4 mL min$^{-1}$, was introduced like the reference air into the ion source (or waste line) of the mass spectrometer, but through the line for sample air. We analysed the standard air about once per two months. Details of the continuous measurement system we used have been reported by Ishidoya and Murayama (2014).

We repeatedly conducted alternate analyses of the sample and reference air, for the continuous measurements of stable isotopic ratios of $O_2$, $N_2$, and Ar ($\delta_{atm}(^{18}O)$, $\delta_{atm}(^{15}N)$, and $\delta_{atm}(^{40}Ar)$) as well as the $O_2/N_2$ ratio and amount fraction of $CO_2$. The time required to obtain a measured value was 62 s. However, the standard deviation of the $\delta_{atm}(^{18}O)$ was about 20 per meg, which was much larger than the standard deviation required to detect the expected respective seasonal (2 per meg) and secular changes ($-0.5$ per meg a$^{-1}$) in $\delta_{atm}(^{18}O)$ calculated by Keeling (1995) and Seibt et al. (2005). We therefore averaged more than 1000 data and used the averaged value as the observed $\delta_{atm}(^{18}O)$. This averaging results theoretically in a standard error of the observed $\delta_{atm}(^{18}O)$ of less than 0.6 per meg assuming no temporal drift during the averaging period. In this regard, we confirmed that the measured values of the $\delta_{atm}(^{18}O)$ against reference air were stable enough for a period much longer than the averaging period. We therefore needed to calibrate with a primary and secondary air standard (described below) only once per two months. Figure 1 shows an example of the measured $\delta_{atm}(^{18}O)$ of a standard air against reference air. We found the standard deviations of 200 and 400 averaged data to be 1.4 and 0.4 per meg, respectively, which are consistent with the theoretically expected values of 1.3 and 1.0 per meg. In general, mass spectrometers behavior can change suddenly due to maintenance, such as filament change, ion source tuning etc.. To minimize the uncertainties associated with the changes in the conditions of the mass spectrometer, we used the specific filaments for the measurements of air samples with the atmospheric level amount fraction of $O_2$ supplied by the Thermo Scientific co.. This enabled us to carry out the continuous measurements in the present study for 11 months without exchanging the filament (when we used the original filament supplied for the mass spectrometer, then we needed to exchange it every 3 months). After the exchange of the filament, several weeks are needed to stabilize the condition of the ion source of the mass spectrometer by flowing the sample and reference air, especially for the elemental ratios as $O_2/N_2$, $Ar/N_2$, and $CO_2/N_2$. Once the condition is stabilized, we did not tune the ion source throughout the period using the same filament. Furthermore, the mass spectrometer was dedicated only to the measurements of $\delta_{atm}(^{18}O)$ and related components including those for flask samples (e.g. Ishidoya et al., 2021, 2022) and it was run day and night autonomously to keep the condition of the ion source.

The $\delta_{atm}(^{18}O)$ and $\delta(O_2/N_2)$ were reported in per meg:

$$\delta_{atm}\left(^{18}O\right) = \frac{R_{sample}\left(^{18}O^{16}O/^{16}O^{16}O\right)-R_{standard}\left(^{18}O^{16}O/^{16}O^{16}O\right)}{R_{standard}\left(^{18}O^{16}O/^{16}O^{16}O\right)}, \tag{1}$$

$$\delta\left(O_2/N_2\right) = \frac{R_{sample}\left(^{16}O^{16}O/^{14}N^{14}N\right)-R_{standard}\left(^{16}O^{16}O/^{14}N^{14}N\right)}{R_{standard}\left(^{16}O^{16}O/^{14}N^{14}N\right)}. \tag{2}$$

Here, the subscript "sample" and "standard" indicate the sample air and the standard air, respectively. Because $O_2$ constitutes 0.2093 mol mol$^{-1}$ of air by volume (Aoki et al., 2019), a change of 4.8 per meg of $\delta(O_2/N_2)$ is equivalent to about a change of 1 $\mu$mol mol$^{-1}$. In this study, the $\delta_{atm}(^{18}O)$ and $\delta(O_2/N_2)$ of each air sample were determined against our primary standard air (cylinder no. CRC00045) using a mass spectrometer. Our standards were dried ambient air or industrially purified air-based

CO$_2$ in 48-L high-pressure aluminium cylinders. The standards were classified as either primary or secondary. Figure 2 shows the value of each analysis and the corresponding annual average of $\delta_{atm}(^{18}O)$ of three secondary standards against the primary air standard. As shown in Fig. 2, variations of the annual average $\delta_{atm}(^{18}O)$ of our three secondary standards were within $\pm 0.8$ to $\pm 1.1$ per meg ($\pm 0.9$ per meg, on average) and nearly stable for 10 years with respect to the primary standard. We therefore allowed an uncertainty of $\pm 0.9$ per meg associated with the stability of the standard air for the annual average $\delta_{atm}(^{18}O)$ in this study. This uncertainty corresponds to an uncertainty of $\pm 0.13$ ($\pm\sqrt{(0.9)^2 + (0.9)^2}/10$) per meg a$^{-1}$ for the 10-year-long secular trend.

We have examined the influence of the amount fraction of CO$_2$ in sample air on the $\delta(O_2/N_2)$ measured on a mass spectrometer in past studies (Ishidoya et al., 2003; Ishidoya and Murayama, 2014). In this study, we also experimentally examined the influences of the amount fraction of CO$_2$ and the $\delta(O_2/N_2)$ on $\delta_{atm}(^{18}O)$. Figure 3a shows typical examples of the relationships between the measured $\delta_{atm}(^{18}O)$ of the air sample and the amount fraction of CO$_2$. To obtain these relationships, a small amount of pure CO$_2$ was added to the flow line of the continuous measurement system during the analysis of standard air, or 1-L flasks were analysed before and after a small amount of pure CO$_2$ was added to the flasks. The precision of the measurements of the flask air samples was about $\pm 4$ per meg. As seen in Fig. 3a, $\delta_{atm}(^{18}O)$ increased linearly with increasing amount fractions of CO$_2$. We therefore decided to correct the $\delta_{atm}(^{18}O)$ values by using amount fractions of CO$_2$ that were measured simultaneously. The mechanism of the positive correlation between the $\delta_{atm}(^{18}O)$ and CO$_2$ was not clarified yet since there is no isobaric interference. In this regard, we found no significant influences of CO$_2$ amount fraction on $\delta_{atm}(^{18}O)$ for a different mass spectrometer, Finnigan MAT-252 (Ishidoya, 2003). This suggests that the influences should be examined carefully for each mass spectrometer. Figure 3b shows the relationships between the measured $\delta_{atm}(^{18}O)$ values of the air samples and their $\delta(O_2/N_2)$. To obtain these relationships, $\delta_{atm}(^{18}O)$ and $\delta(O_2/N_2)$ were measured for 1-L flasks or 48-L cylinders before and after pure N$_2$ was added to them. One-litre flasks were filled with the air in the cylinders for the analyses. It is apparent from Fig. 3b that we did not find a clearly increasing or decreasing trend of $\delta_{atm}(^{18}O)$, at least when the $\delta(O_2/N_2)$ was decreasing by about $-8000$ per meg. We therefore decided that we would not correct the $\delta_{atm}(^{18}O)$ values for the changes of the simultaneously measured $\delta(O_2/N_2)$. It is noteworthy that we obtained a different result—an increase of $\delta_{atm}(^{18}O)$ with a decrease of $\delta(O_2/N_2)$— in our earlier flask studies in 2013. We have not yet clarified the cause(s), but we expect the results shown in Fig. 3b are valid because we repeatedly obtained results that were consistent between flasks and cylinders.

## 2.2 Box model for simultaneous analysis of $\delta_{atm}(^{18}O)$ and $\delta(O_2/N_2)$

The box model used in this study is the same as that described by B94. However, the isotopic effects for the sink/source processes are updated by more recent study of L&B11. Therefore, we performed calculations using the isotope effects from both studies and compared them especially for the long-term variations of $\delta_{atm}(^{18}O)$. Since there are a lot of symbols used in

this study, we present a list of the symbols in the main text in Table A1 in Appendix A. Equation (3) is the mass balance equation for $\delta_{atm}(^{18}O)$ (See Appendix B for derivation).

$$\frac{d\delta_{atm}(^{18}O)}{dt} = (\varepsilon_{MR}r_{MR} + \varepsilon_{PR}r_{PR} + \varepsilon_{DR}r_{DR})R_{Res} + (\Delta_{PS} - \varepsilon_{LE} - \delta_{atm})R_{PS}$$

$$+\varepsilon_{OR}R_{OR} + (\Delta_{OW} - \delta_{atm})R_{OP} + \varepsilon_{TS}R_{TS} + \varepsilon_{ST}R_{ST} + \varepsilon_{FF}R_{FF}, \qquad (3)$$

where $r_{MR}$, $r_{PR}$, and $r_{DR}$ are the relative ratios of the Mehler reaction, photorespiration, and dark respiration to the total $O_2$ consumption associated with terrestrial respiration, and $\varepsilon_{MR}$, $\varepsilon_{PR}$, $\varepsilon_{DR}$, $\varepsilon_{LE}$, and $\varepsilon_{OR}$ denote the isotope effects of the Mehler reaction, photorespiration, dark respiration, leaf water enrichment, and marine respiration, respectively. Values of $O_2$ budgets and isotopic effects are summarized in Table 1. The isotopic enrichment of $O_2$ produced by the terrestrial photosynthesis, $\Delta_{PS}$, is basically determined by the $\delta_{LW}(^{18}O)$ (Gonfiantini et al.,1965; Dongmann et al., 1972; Farquhar et al., 1993), and we assumed $\Delta_{PS}$, to be 4.4 or 6.5 ‰ for the steady state. There is still a large uncertainty in $\delta_{LW}(^{18}O)$ (Farquhar et al., 1993; Bender et al., 1994; Hoffmann et al., 2004; Keeling 1995; West et al., 2008). In this study, we used the $\delta_{LW}(^{18}O)$ calculated by a 3-D model (see Sect. 3.2 and 3.3). We used values of the relative ratios for the respirations ($r_{MR}$, $r_{PR}$, and $r_{DR}$) following B94. Specifically, respective values of $r_{MR}$, $r_{PR}$, and $r_{DR}$ are 0.1, 0.31, and 0.59 in B94. As a result, the total isotope effects of terrestrial respiration become 18.7 and 17.7 ‰ for B94 and L&B11, respectively. This difference is caused by the large difference of their dark respiration effects ($\varepsilon_{DR}$). The biggest difference between the two studies is the effect of ocean, and the respective oceanic DME are 18.9 and 23.5 ‰ for B94 and L&B11. The oceanic DME for L&B11 is almost the same magnitude as the total terrestrial DME, which are 22.4 (4.4 – 0.70 + 18.7) and 23.5 (6.5 – 0.75 + 17.7) ‰ for B94 and L&B11, respectively. L&B11 also showed that photosynthetic enrichment in the ocean cannot be ignored, contrary to the previous studies. However, they did not clearly separate the effects of the oceanic photosynthesis and respiration. Therefore, we assumed $\Delta_{OW}$ and $\varepsilon_{OR}$ to be zero and 23.5 ‰, respectively, to set the total oceanic DME to be 23.5 ‰ as L&B11. Unlike the previous studies, the stratospheric effect was formulated as fractionations, $\varepsilon_{TS}$ and $\varepsilon_{ST}$, which denote the isotope effects of air exchange between the troposphere and stratosphere. This is because we have continued precise measurements of the isotopic ratios of $O_2$ in the stratosphere, which could provide new insights into stratospheric processes, as described later. $R_{Res}$, $R_{PS}$, $R_{OR}$, and $R_{OP}$ (the unit is $a^{-1}$) represent the relative ratios of the annual fluxes of $O_2$ from terrestrial respiration, terrestrial production, marine respiration, and marine production, respectively, to the total amount of $O_2$ in the atmosphere (=3.706 x $10^4$ Pmol). For example, if we assume that the terrestrial flux is 16.7 Pmol $a^{-1}$, $R_{PS}$ will be 16.7/(3.706 x $10^4$) = 4.5 x $10^{-4}$ $a^{-1}$, as shown in Table 1. $R_{TS}$ and $R_{ST}$ denote the relative ratios of the annual fluxes of $O_2$ between the troposphere and stratosphere, respectively. $\varepsilon_{FF}$ and $R_{FF}$ denote the isotopic effects in fossil fuel combustion and the relative ratios of the annual $O_2$ consumption by fossil fuel combustion, respectively. We assumed that atmospheric oxygen is consumed without isotope effects in fossil fuel combustion ($\varepsilon_{FF}=0$), taking into account that the industrial combustion processes usually occur at high temperature. Therefore, we consider

no contribution to DME from fossil fuel combustion in this study. In this regard, it is known that large oxygen isotope fractionation occurs in the combustion processes such as biomass burning due to complex combustion processes (Schumacher et al., 2011). In such cases, it will be necessary to consider isotopic fractionation in the consumption of atmospheric oxygen associated with combustion. However, at present, little is known about the impact of this on DME. The box model also calculates the amount fraction of atmospheric $O_2$, $y(O_2)$ by solving the following mass balance equation:

$$\frac{1}{y(O_2)}\frac{dy(O_2)}{dt} = (r_{MR} + r_{PR} + r_{DR})R_{Res} + R_{PS} + R_{OR} + R_{OP} + R_{TS} + R_{ST} + R_{FF} \tag{4}$$

Here, $y$ stands for the dry amount fraction of gas, as recommended by the IUPAC Green Book (Cohen et al., 2007). To compare with the observed results for $\delta(O_2/N_2)$, the amount fraction of $O_2$ calculated by the box model was converted to $\delta(O_2/N_2)$ assuming a norm atmosphere.

We assumed the value of terrestrial $O_2$ production, $P_T$, to be 16.7 Pmol a$^{-1}$, which is the value reported by Hoffmann et al. (2004). The ratio of the terrestrial and marine productions was assumed to be 0.63:0.37 (Luz and Barkan, 2011). It is known that mass-independent isotopic fractionation of $^{17}O$ and $^{18}O$ between $O_3$ and $CO_2$ occurs in the stratosphere via photochemical processes (e.g., Gamo et al., 1989; Thiemens, 1999). B94 has estimated the isotopic effect on atmospheric $O_2$ by scaling the $\delta(^{18}O)$ of $CO_2$ and calculated that it would depress $\delta_{atm}(^{18}O)$ by 0.4 ‰, considering the turnover time between the troposphere and the stratosphere. L&B11 has shown that the global $\Delta(^{17}O)$ budget supports their result and have estimated the stratospheric isotope effect on $\delta_{atm}(^{18}O)$ to be 0.3 ‰. In this study, the flux between the troposphere and stratosphere was set to 3000 Pmol a$^{-1}$, which is calculated from the stratosphere-troposphere (S-T) mass flux (Olesen et al., 2004). This S-T $O_2$ flux is approximately 100 times the flux from the biosphere (Luz et al., 1999). $\varepsilon_{ST}$, which is the isotopic fractionation of $O_2$ that returns from the stratosphere to the troposphere, is currently considered to be so small that it is impossible to actually detect it in the stratosphere. Note that $\varepsilon_{ST}$ represents the fractionation of stratosphere-troposphere exchange flux in this study, while $\varepsilon_{strat}$ in L&B11 represents the stratospheric effect on DME. As a rough estimate, considering that the value of $\Delta(^{17}O)$ is –1.5 per meg with respect to the tropospheric value (Luz et al., 1999), $\varepsilon_{ST}$ is expected to be about –3 per meg based on the mass independent effect ($\delta^{17}O \approx \delta^{18}O$). Here, $\varepsilon_{ST}$ was set to –2.5 per meg so that the diminution of $\delta_{atm}(^{18}O)$ at equilibrium was –0.3 ‰. Because there are no isotopic effects during the transport of air from the troposphere to the stratosphere, $\varepsilon_{TS}$ should be zero.

Based on the above discussion, the $\delta(^{18}O)$ in the stratosphere should be –2.5 to –3 per meg lower than in the troposphere due to the photochemical processes. As a matter of fact, the $\delta(^{18}O)$ of stratospheric $O_2$ has been observed with high precision by balloon experiments and is known to decrease significantly with increasing altitude because of gravitational separation (e.g., Ishidoya et al., 2013b; Sugawara et al., 2018). At an altitude of 35 km over Japan, $\delta(^{18}O)$ is lower than the tropospheric value by approximately –100 per meg, which is anomalously lower (i.e. larger diminution) than that expected on the basis of photochemical diminution. The implication is that enrichment of approximately 5 per meg is permanently occurring in the

troposphere because of gravitational separation in the stratosphere (Ishidoya et al., 2021). It is currently uncertain how gravitational separation affects the process by which isotopically light oxygen is transported to the troposphere through troposphere–stratosphere exchange. For example, changing $\varepsilon_{ST}$ from –2.5 to –5.0 per meg yields the $\delta_{atm}(^{18}O)$ trend of approximately –0.2 per meg a$^{-1}$, because the flux from the stratosphere is over 100 times greater than the surface biospheric flux. This uncertainty complicates the problem of inter-annual $\delta_{atm}(^{18}O)$ change and suggests that gravitational separation may be involved in small fluctuations in the DME.

With these initial settings, we were able to reach a steady state after a 5000-year simulation, and we found that the equilibrium value of $\delta_{atm}(^{18}O)$ were 20.82 and 23.16 ‰ for calculations following B94 and L&B11, respectively, which are almost same values reported by two studies. Hereafter, the box model results are discussed based on the differences from these equilibrium value. The biospheric turnover time of $O_2$ in the steady state was 1398 years, which is longer than the 1200 years estimated by B94. This may be a little too long, since the $\delta_{atm}(^{18}O)$ variations reported by Severinghaus et al. (2009) based on the ice core measurements that showed a characteristic asymptotic decay curve after abrupt climate change events on a timescale of about ~1000 years, implying that the turnover time of $O_2$ in the atmosphere is about 1000 years. The biospheric turnover time is inversely proportional to the sum of the terrestrial and oceanic productions of $O_2$ incorporated into the box model, which is 26.5 (16.7 + 9.8) Pmol a$^{-1}$ in this study (Table 1). This implies that total production of $O_2$ for the initial value in our model is underestimated. In this regard, turnover time decreases to about 1000 years when we simulate a case in which the GPP is increased, as will be discussed later. In model calculations for the interpretation of long-term changes, we used the steady-state condition described above as the initial condition, and we performed some calculations by adding long-term changes to terrestrial GPP, photorespiration, and $\delta_{LW}(^{18}O)$ (see details in Sect. 3.3).

The box model was suitable for simulations if we assumed that long-term and global changes occurred over timeframes of hundreds to thousands of years. The box model naturally ignores atmospheric transport processes, and it is difficult to define the box atmosphere at local and regional spatial scales. There is hence a theoretical limit to the application of the box model to short-timescale phenomena. However, we tried to use this box model as a first step to understand the diurnal and seasonal changes of $\delta_{atm}(^{18}O)$ recently observed by high-precision measurements. Because $\delta(O_2/N_2)$ was also observed at the same time during this study, the relationships between $\delta_{atm}(^{18}O)$ and $\delta(O_2/N_2)$ provided information about the usefulness of the box model simulations. For the simulations of diurnal changes, the intensities of terrestrial $O_2$ consumption and production were approximated by a simple function, which became a maximum at noon and zero during the night. We also carried out simulations of diurnal changes considering marine $O_2$ consumption and production approximated by the similar simple function, to examine sensitivities of $\delta_{atm}(^{18}O)$ / $\delta(O_2/N_2)$ ratio to the terrestrial and marine signals. Seasonal variations were also simulated by a simple sinusoidal function. We then tuned the magnitude of $R_{Res}$ (=$R_{PS}$) so that the amplitude of the modelled $\delta(O_2/N_2)$ variation was close to the observed results. The box model did not incorporate the contributions of S-T $O_2$ flux and fossil fuel combustion for the simulations of the diurnal and seasonal changes.

## 2.3 Numerical simulations of $\delta_{LW}(^{18}O)$ using the 3-D model MIROC5-iso

We simulated $\delta_{LW}(^{18}O)$ using a stable water isotope-enabled general circulation model named MIROC5-iso (Okazaki and Yoshimura, 2017, 2019). MIROC5-iso is the fifth generation of the Model for Interdisciplinary Research on Climate (MIROC5; Watanabe et al., 2010). The stable water isotopes were implemented to the atmospheric and land-surface components following Jouzel et al. (1987) and Yoshimura et al. (2006). The MIROC5-iso calculates the isotopic ratio of atmospheric water vapor, precipitation, and reservoirs at ground level, including soil water and leaf water, with the equilibrium

and kinetic fractionation at all phase transitions. The $\delta_{LW}(^{18}O)$ is calculated by considering water conveyance driven by transpiration and diffusive isotopic movement (i.e., "back diffusion") as follows:

$$\frac{dV_L}{dz_L}\frac{\partial R_{LW}}{\partial t} = \frac{1}{\rho}\left(\frac{TA_L}{I_{LA}}\frac{\partial R_{LW}}{\partial z_L} - D\tau\frac{\partial^2 R_{LW}}{\partial z_L^2}\right). \tag{5}$$

Here, $R_{LW}$ is the isotopic ratio of the leaf water given by $R_{LW} = R_{sample}\left(H_2{}^{18}O/H_2{}^{16}O\right)/R_{standard}\left(H_2{}^{18}O/H_2{}^{16}O\right)$, where the subscript "sample" and "standard" indicate the sample and the standard water, respectively, and the standard water is V-

SMOW. $z_L$ is the axis directed from leaf base to tip, and $V_L$ and $A_L$ are the volume of leaf water and area of leaf surface, respectively. $\rho$ is the density of water, $T$ is the transpiration flux, $I_{LA}$ is the leaf area index, $D$ is the liquid diffusivity of an isotope, and $\tau$ is the crookedness of the leaf. The transpired water drawn up from the root zone layers is calculated by weighting the isotope ratio of soil water by root density. The transpiration fluxes of the water isotopes were calculated by the bulk method with the bulk exchange coefficient of Sellers et al. (1996), and the equilibrium and kinetic fractionations from liquid to gas at

the stoma were considered.

In this study, MIROC5-iso was forced by observed sea surface temperature, sea ice concentration, observed greenhouse gases (carbon dioxide, methane, and chlorofluorocarbons), ozone, and changes of land use. The isotopic compositions of sea surface water and sea ice were kept constant and assumed to be 0 ‰ and 3 ‰, respectively, as in Joussaume and Jouzel (1993). The model resolution was set to T42 (approximately 280 km at the equator) with 40 vertical levels. After running MIROC5-iso for

100 years with the condition of AD 1871 for spin-up, we ran the model for AD 1871–2022.

## 3 Results and Discussion

## 3.1 Diurnal variations of $\delta_{atm}(^{18}O)$ and $\delta(O_2/N_2)$

265 Figure 4a shows the average diurnal cycles of $\delta_{atm}(^{18}O)$, $\delta(O_2/N_2)$, the amount fraction of $CO_2$, and $\delta(Ar/N_2)$ for each season observed at TKB during 2013–2022. $\delta(Ar/N_2)$ was defined in the same way as $\delta(O_2/N_2)$ but for the $^{40}Ar/^{14}N^{14}N$ ratio. The error bands shown in Fig. 4a indicate year-to-year variations of the average diurnal cycles ($\pm 1\sigma$). In this study, we needed to remove any natural or artificial fractionation of $^{18}O^{16}O$ and $^{16}O^{16}O$, other than the processes associated with the DME from the observed $\delta_{atm}(^{18}O)$. For this purpose, we used the observed diurnal $\delta(Ar/N_2)$ cycle, which is potentially driven by the night-time vertical

270 temperature gradient (Adachi et al., 2006) and artificial inlet fractionation induced by radiative heating of an air intake (e.g., Blaine et al., 2006). The $\delta(Ar/N_2)$ underwent a slight diurnal cycle with a maximum in the early morning (Fig. 4a), and the difference between the maximum and minimum was about 4–6 per meg. It is difficult to specify the cause of the diurnal cycle, but it may result from natural variations due to a night-time vertical temperature gradient at the inland TKB site, because Adachi et al. (2006) have reported much larger enrichment of $\delta(Ar/N_2)$ by 100 per meg at the centre of a wide desert during

275 the night. We therefore decided to correct the observed values for thermally diffusive fractionation following the method used by Ishidoya et al. (2014, 2022) and to use the corrected values for our discussion of diurnal variations. Specifically, we subtracted $(1.55/16.2) \times \delta(Ar/N_2)$ from the observed $\delta_{atm}(^{18}O)$. The coefficients 1.55 and 16.2 are the $\delta_{atm}(^{18}O) / \delta(Ar/N_2)$ ratios determined by laboratory experiments (Ishidoya et al., 2013b). In a similar manner, we also corrected the $\delta(O_2/N_2)$ for thermally diffusive fractionation by subtracting $(4.57/16.2) \times \delta(Ar/N_2)$ from the measured $\delta(O_2/N_2)$. The coefficient

280 $4.57/16.2$ is the $\delta(O_2/N_2) / \delta(Ar/N_2)$ ratio from the same laboratory experiments. The maximum corrections were 0.3 and 1.0 per meg for $\delta_{atm}(^{18}O)$ and $\delta(O_2/N_2)$, respectively. The correction for the amount fraction of $CO_2$ was negligibly small.

 $\delta_{atm}(^{18}O)$ exhibited a clear diurnal cycle with a daytime minimum, especially in summer (Fig. 4a). $\delta_{atm}(^{18}O)$ varied out of phase with $\delta(O_2/N_2)$, and the ratio of the amplitude of the diurnal $\delta_{atm}(^{18}O)$ cycles to those of $\delta(O_2/N_2)$ was substantially larger in summer than in winter. Figure 4a also shows the diurnal cycles of $\delta_{atm}(^{18}O)$ and $\delta(O_2/N_2)$ simulated by the box model described

285 in Sect. 2.2 that incorporated the isotopic effects from B94. The simulations were carried out under two conditions, one was the case when we ignored marine respiration and the production of $O_2$ ($R_{OR}$ and $R_{OP}$), and the other was the case when we ignored terrestrial respiration and the production of $O_2$ ($R_{Res}$ and $R_{PS}$). In both cases, the $R_{Res}$ and $R_{PS}$ (or $R_{OR}$ and $R_{OP}$) in the model were adjusted to reproduce the observed seasonal $\delta(O_2/N_2)$ cycle subject to the constraint that the daily average $R_{Res} = R_{PS}$ (or $R_{OR} = R_{OP}$). The initial value of the $\delta_{atm}(^{18}O)$ relative to ocean water was then adjusted arbitrarily to establish a steady

290 state for the simulated $\delta_{atm}(^{18}O)$. The $\delta(^{18}O)$ values relative to ocean water in the steady state were 22.0 and 18.9 ‰ for the case when we considered only terrestrial or only marine respiration/production, respectively. We found that the general characteristics of the diurnal cycles of $\delta_{atm}(^{18}O)$ and $\delta(O_2/N_2)$ were reproduced by the simulated $\delta_{atm}(^{18}O)$ for both cases when we considered only terrestrial or only marine processes (Fig. 4a).

To determine the cause(s) of the observed diurnal $\delta_{atm}(^{18}O)$ cycles, we examined the relationships between the observed $\delta_{atm}(^{18}O)$ and $\delta(O_2/N_2)$ (Fig. 4b), and those for $\delta(O_2/N_2)$ and the amount fraction of $CO_2$ (Fig. 4c). The $\delta_{atm}(^{18}O)$ / $\delta(O_2/N_2)$ ratios were −0.017 and –0.006 per meg (per meg)$^{-1}$ in summer and winter, respectively. In Fig. 4b, we also plot the relationship between the simulated $\delta_{atm}(^{18}O)$ and $\delta(O_2/N_2)$. We found the simulated ratios to be –0.017 and –0.019 per meg (per meg)$^{-1}$ when we considered only terrestrial or only marine respiration/production, respectively. These ratios were much closer to the ratio observed in summer than in winter. The $O_2$ and $CO_2$ exchange ratios (ER, $-\Delta y(O_2)\Delta y(CO_2)^{-1}$) calculated from the $\delta(O_2/N_2)$ and the amount fraction of $CO_2$ shown in Figure 4c, were 1.08 and 1.45 in summer and winter, respectively. An oxidative ratio (OR, $-\Delta y(O_2)\Delta y(CO_2)^{-1}$) of 1.05–1.1 is expected for terrestrial biosphere activities, and ratios of 1.17, 1.44, and 1.95 are expected for combustion of solid fuel, liquid fuel, and natural gas, respectively (Keeling, 1988; Severinghaus, 1995). The ER refers to the exchange between the atmosphere and organisms or ecosystems, whereas the OR reflects the stoichiometry of specific materials, in accord with Faassen et al. (2023) and Ishidoya et al. (2024). The ORs therefore suggested that the diurnal $\delta(O_2/N_2)$ cycle observed in summer could be attributed mainly to terrestrial biosphere activities, whereas that in winter was due to fossil fuel combustion. The observed wintertime ER of 1.45 was also consistent with the average OR of $1.52 \pm 0.1$ for fossil fuel consumption (hereafter referred to as "OR$_{FF}$") for the Kanto area, which includes TKB, of about $1.7 \times 10^4$ km$^2$, calculated using the data on fossil fuel consumption reported by the Agency of Natural Resources and Energy (https://www.enecho.meti.go.jp/statistics/energy_consumption/ec002/results.html#headline2, last access: 28 March 2024, in Japanese) (Ishidoya et al., 2020). The implication is therefore that the isotopic discrimination of $O_2$ during activities of the terrestrial biosphere was the main cause of the observed summertime diurnal $\delta_{atm}(^{18}O)$ and $\delta(O_2/N_2)$ cycles, and the isotopic discrimination of $O_2$ during fossil fuel combustion was very small or negligible.

The simulated diurnal cycle of $\delta_{atm}(^{18}O)$ and the $\delta_{atm}(^{18}O)$ / $\delta(O_2/N_2)$ ratio for the case when only terrestrial processes were considered were very similar to those for the case when only marine processes were considered. This similarity was due to the small difference between the isotopic discriminations of the terrestrial and marine processes (22.4 – 18.9 = 3.5 ‰). If we use the isotopic discriminations from L&B11, then the corresponding difference is much smaller (23.5 – 23.5 = 0 ‰). We could therefore estimate the variations of the observed $\delta(O_2/N_2)$ driven by the total activities of the terrestrial and marine biosphere (hereafter referred to as "$\delta_{BIO}(O_2/N_2)$") by dividing the observed variations in $\delta_{atm}(^{18}O)$ by the ratio of the simulated $\delta_{atm}(^{18}O)$ / $\delta(O_2/N_2)$ of about –0.017 to –0.019 per meg (per meg)$^{-1}$. We could then estimate the variations of $\delta(O_2/N_2)$ driven by fossil fuel combustion (hereafter referred to as "$\delta_{FF}(O_2/N_2)$") by subtracting the $\delta_{BIO}(O_2/N_2)$ from the observed $\delta(O_2/N_2)$. This method, hereafter referred to as the "$\delta_{atm}(^{18}O)$-method", enabled us to remove the impact on $\delta(O_2/N_2)$ of not only the activities of the terrestrial biosphere but also the contributions due to the air–sea $O_2$ flux, which is driven mainly by activities in the marine biosphere (e.g., Nevison et al., 2012; Eddebbar et al., 2017), from the estimated $\delta_{FF}(O_2/N_2)$. For an application of the $\delta_{atm}(^{18}O)$-method, we assume there is no isotopic discriminations during fossil fuel combustion considering the seasonal differences in the $\delta_{atm}(^{18}O)$ / $\delta(O_2/N_2)$ ratios in Fig. 4b. It would be generally reasonable since the combustion occurs at high temperature,

which minimizes isotopic discriminations. However, Schumacher et al. (2011) reported isotopic discriminations on the order of up to 26 ‰ for stable oxygen isotopic ratio of atmospheric $CO_2$ ($\delta_{CO2}(^{18}O)$) derived from combustion of different kinds of material. They suggested that natural combustion processes on the long term might enrich $\delta_{atm}(^{18}O)$ and contribute to the DME. Therefore, isotopic discriminations of $\delta_{atm}(^{18}O)$ due to combustion processes should be examined carefully in future, based on precise observations of $\delta_{atm}(^{18}O)$.

Figure 5 shows the $\delta_{BIO}(O_2/N_2)$ and $\delta_{FF}(O_2/N_2)$ estimated by the $\delta_{atm}(^{18}O)$-method for each season. The largest amplitudes of the diurnal $\delta_{BIO}(O_2/N_2)$ and $\delta_{FF}(O_2/N_2)$ cycles were in summer and winter, respectively. For comparison, we separated the contributions of terrestrial biosphere activities and fossil fuel combustion to the observed $\delta(O_2/N_2)$ based on the observed ER and amount fraction of $CO_2$ (hereafter referred to as "ER-method"). For this purpose, (1) we assumed that the diurnal cycle of the amount fraction of $CO_2$ was driven by terrestrial biosphere activities and fossil fuel combustion, (2) we ignored the contribution of the air–sea $O_2$ flux on $\delta(O_2/N_2)$, and (3) we assumed the OR for activities in the terrestrial biosphere ($OR_B$) to be 1.1 (Severinghaus, 1995), which has been widely used in past studies (e.g. Manning and Keeling, 2006; Tohjima et al.), and the $OR_{FF}$ to be 1.4, 1.5, 1.6, or 1.7 considering mixed combustion of solid fuel, liquid fuel, and natural gas. It is noted some recent studies have used the $OR_B$ of 1.05 rather than 1.1 (e.g. Morgan et al., 2021).

The equations for the ER-method can be written as:

$$\Delta y(CO_2, B) + \Delta y(CO_2, FF) = \Delta y(CO_2), \qquad (6)$$

$$\frac{\Delta y(CO_2,B) \times \alpha_B + \Delta y(CO_2,FF) \times \alpha_F}{\Delta y(CO_2,B) + \Delta y(CO_2,FF)} = \alpha_{obs}. \quad (7)$$

Here, $\Delta y(CO_2, B)$ and $\Delta y(CO_2, FF)$ are changes in the amount fraction of $CO_2$ driven by terrestrial biosphere activities and fossil fuel combustion, respectively. $\Delta y(CO_2)$ is the observed average diurnal cycle of the amount fraction of $CO_2$ for each season shown in Fig. 4a. $\alpha_B$, $\alpha_F$, and $\alpha_{obs}$ are the $OR_B$, $OR_{FF}$, and the observed ER for each season shown in Fig. 4c. Once $\Delta y(CO_2, B)$ and $\Delta y(CO_2, FF)$ are calculated by solving eqs. (6) and (7), they can be converted to $\Delta\delta(O_2/N_2)$ by using $OR_B$ and $OR_{FF}$, respectively.

Figure 5 shows the $\delta(O_2/N_2)$ driven by terrestrial biosphere activities and fossil fuel combustion estimated by the ER-method. The results agreed well with the $\delta_{BIO}(O_2/N_2)$ and $\delta_{FF}(O_2/N_2)$ for all seasons, especially when we chose the $OR_{FF}$ to be 1.6 or 1.7, which are higher and lower than those expected from liquid fuel and natural gas fuel combustion, respectively. The implication is therefore that the diurnal $\delta_{FF}(O_2/N_2)$ cycles at TKB were driven by car traffic (liquid fuels) and household gas consumption. It is noteworthy that propane ($CH_3CH_2CH_3$), for which the $OR_{FF}$ is 1.67 assuming complete combustion, should also be considered as the household gas consumed in the TKB area.

To determine whether variations of $\delta_{atm}(^{18}O)$ on hourly to daily timeframes were observable, we plotted typical examples in Fig. 6 of rolling averages calculated over 300 cycles (5 hours) and 1100 cycles (19 hours) of mass spectrometric measurements of $\delta_{atm}(^{18}O)$ and $\delta(O_2/N_2)$. The summertime graphs (Fig. 6a) clearly showed that the $\delta_{atm}(^{18}O)$ varied in antiphase with $\delta(O_2/N_2)$ on timescales of both 5 and 19 hours. The ratios of $\delta_{atm}(^{18}O)$ / $\delta(O_2/N_2)$ were $-0.017$ per meg (per meg)$^{-1}$ for data averaged over both 5 h and 19 h. This result agreed with that obtained from the summertime average diurnal cycle (vide supra). In contrast, there was no clear correlation between variations of $\delta_{atm}(^{18}O)$ and $\delta(O_2/N_2)$ in winter (Fig. 6b). We could distinguish some short-term $\delta_{atm}(^{18}O)$ variations (Fig. 6b), but the causes were unclear. The variations may be partly due to activities in the biosphere, because the $\delta_{atm}(^{18}O)$ in winter showed small but substantial diurnal cycles of $\delta_{atm}(^{18}O)$ and $\delta_{BIO}(O_2/N_2)$ (Figs. 4a and 5a). These characteristics suggest that we could apply the $\delta_{atm}(^{18}O)$-method to resolve temporal variations of $\delta_{BIO}(O_2/N_2)$ and $\delta_{FF}(O_2/N_2)$ separately on timeframes of several hours to day-to-day. Similar separation has been carried out for $CO_2$ based on the simultaneous analysis of the $\Delta(^{14}C)$ and amount fraction of $CO_2$ (e.g., Graven et al., 2018; Basu et al., 2016, 2020) or based on the simultaneous analysis of $\delta(O_2/N_2)$ and the amount fraction of $CO_2$ by assuming an average $OR_{FF}$ based on a statistical assessment (e.g. Minejima et al., 2012; Sugawara et al., 2021; Pickers et al., 2022). The $\delta_{atm}(^{18}O)$-method may have some advantages compared with methods used in previous studies because we could apply it without assuming any $OR_{FF}$ with a temporal resolution of 5 hours or perhaps even shorter.

## 3.2 Seasonal cycles of $\delta_{atm}(^{18}O)$ and $\delta(O_2/N_2)$

Figure 7a shows the monthly mean values of $\delta_{atm}(^{18}O)$ and $\delta(O_2/N_2)$ at TKB during 2013–2022. To reduce local effects of fossil fuel combustion around TKB, we extracted the successive maxima of $\delta(O_2/N_2)$ for 4320 cycles (3 days) of mass spectrometric measurements to calculate the monthly mean values of $\delta(O_2/N_2)$ plotted in Fig. 7a. In contrast, all data were used to calculate the monthly mean values of $\delta_{atm}(^{18}O)$ to reduce their standard errors, because fossil fuel combustion did not change $\delta_{atm}(^{18}O)$ significantly, as discussed in section 3.1. We removed anomalous $\delta_{atm}(^{18}O)$ data from the plot during four months when the mass spectrometer was producing unreliable results. Therefore, Fig. 7a shows 116 and 120 data of $\delta_{atm}(^{18}O)$ and $\delta(O_2/N_2)$, respectively. Some seasonal and interannual variations are apparent in Fig. 7a, not only for $\delta(O_2/N_2)$, which has been reported in many past studies (Keeling and Manning, 2014), but also for $\delta_{atm}(^{18}O)$. We examined the observed average seasonal cycle and secular trend of $\delta_{atm}(^{18}O)$, and in the following paragraphs we discuss the implications thereof for the oxygen, carbon, and water cycles.

Figure 7b shows the average seasonal cycles of $\delta_{atm}(^{18}O)$ and $\delta(O_2/N_2)$ at TKB during 2013–2022. The $\delta(O_2/N_2)$ values in this figure are the values after contributions from the solubility changes in the ocean were removed. For this purpose, we used the seasonal $\delta(Ar/N_2)$ cycle, which is driven mainly by the air–sea heat flux at the surface (e.g., Keeling et al., 2004; Ishidoya et

al., 2021; Morgan et al., 2021). Specifically, the average seasonal cycle of $\delta(Ar/N_2)$ at TKB (Ishidoya et al., 2021), multiplied by a coefficient of 0.9 derived from differences in the solubilities of $O_2$ and Ar (Weiss, 1970), was subtracted from the average seasonal cycle of $\delta(O_2/N_2)$. We applied this correction so that we could discuss the variations of $\delta(O_2/N_2)$ associated with only the DME. The peak-to-peak amplitude of the corrected seasonal $\delta(O_2/N_2)$ cycle was smaller than that of the uncorrected seasonal $\delta(O_2/N_2)$ cycle by about 7 per meg. It is apparent in Fig. 7b that the $\delta_{atm}(^{18}O)$ varied seasonally, roughly in antiphase to the seasonal cycle of $\delta(O_2/N_2)$. The minimum of the seasonal $\delta_{atm}(^{18}O)$ cycle appeared in late summer to early autumn, and the peak-to-peak amplitude was 2.1±0.6 per meg. The maximum of the seasonal $\delta(O_2/N_2)$ cycle occurred in summer, and its peak-to-peak amplitude was 113±10 per meg. The uncertainties for the amplitudes of $\delta_{atm}(^{18}O)$ ($\delta(O_2/N_2)$) was evaluated as a standard deviation of the 10-year average monthly mean values from the best-fit curve shown in Fig. 7b. Keeling (1995) expected $\delta_{atm}(^{18}O)$ to be lower in summer than in winter by 2 per meg based on the assumption that the 100 per meg seasonal increase of $\delta(O_2/N_2)$ was driven by the input of photosynthetic $O_2$, the $\delta(^{18}O)$ of which is about 20 ‰ lower than $\delta_{atm}(^{18}O)$. This can be calculated as:

$$\delta_{atm\_summer}(^{18}O) = \frac{\left(y(O_2)_{winter} \times \delta_{atm\_winter}(^{18}O) + \Delta y(O_2) \times \delta_{LW}(^{18}O)\right)}{y(O_2)_{winter} + \Delta y(O_2)}. \quad (8)$$

Here, $\delta_{atm\_summer}(^{18}O)$ and $\delta_{atm\_winter}(^{18}O)$ are $\delta_{atm}(^{18}O)$ in summer and winter, respectively, and $y(O_2)_{winter}$ is atmospheric $O_2$ amount fraction in winter, and $\Delta y(O_2)$ is an input of photosynthetic $O_2$ to the atmosphere. If we assume $\delta_{atm\_winter}(^{18}O)$ and $\delta_{LW}(^{18}O)$ are 0 and –20 ‰, respectively, $y(O_2)_{winter}$ is 209400 μmol mol$^{-1}$, and $\Delta y(O_2)$ is 21 μmol mol$^{-1}$, which corresponds to 100 per meg seasonal increase of $\delta(O_2/N_2)$, then we obtain $\delta_{atm\_summer}(^{18}O)$ of –2 per meg as Keeling (1995). Although his estimation was relatively simple, it reproduced the general characteristics of the seasonal $\delta_{atm}(^{18}O)$ and $\delta(O_2/N_2)$ cycles observed in the present study well. In Fig. 7b, we also plotted the seasonal cycles of $\delta_{atm}(^{18}O)$ and $\delta(O_2/N_2)$ simulated by our box model that incorporated the isotopic effects from B94. The $R_{Res}$, $R_{PS}$, $R_{OR}$, and $R_{OP}$ values in the model were adjusted to reproduce the observed seasonal $\delta(O_2/N_2)$ cycle by imposing the constraints that the annual average $R_{Res} = -R_{PS}$ and $R_{OR} = -R_{OP}$. We then adjusted the initial value of the $\delta_{atm}(^{18}O)$ in the model arbitrarily to establish a steady state for the simulated $\delta_{atm}(^{18}O)$. We set the $R_{Res} / R_{OR}$ (or $R_{PS} / R_{OP}$) ratio to be 2 for the simulation in Fig. 7b. As discussed in Sect. 3.1, changes of that ratio do not substantially change the simulated results of $\delta_{atm}(^{18}O)$. Figure 7c shows the same simulated seasonal cycles of $\delta_{atm}(^{18}O)$ and $\delta(O_2/N_2)$ in Fig. 7b, and the respective contributions of terrestrial production, terrestrial respiration, ocean production, and ocean respiration. As seen from Fig. 7c, seasonal $\delta_{atm}(^{18}O)$ cycle is driven mainly by production rather than respiration, which is consistent with the estimation by Keeling (1995). In this context, seasonal cycles of $\delta_{CO2}(^{18}O)$ have been reported by some past studies (e.g. Peylin et al., 1999; Cuntz et al., 2003; Murayama et al., 2010). Peylin et al. (1999) and Cuntz et al. (2003) used 3-D atmospheric transport models to reproduce the observations, and they found the main contributors are respiration and production for the respective seasonal cycles of $\delta_{CO2}(^{18}O)$ and $CO_2$ amount fraction. These characteristics

are different from the seasonal cycles of $\delta_{atm}(^{18}O)$ and $\delta(O_2/N_2)$ observed in this study, both of which are driven mainly by production (Fig. 7c).

We found that the box model could reproduce the observed seasonal $\delta_{atm}(^{18}O)$ cycles, although both the seasonal minimum and maximum of the simulated seasonal $\delta_{atm}(^{18}O)$ cycle appeared slightly earlier (by about 1 month) than in the observed cycle represented by a one-harmonic, best-fit curve (Fig. 7b). One- to two-month time shift between the observed and simulated seasonal cycles are also found in $\delta_{CO2}(^{18}O)$ at various surface stations (Peylin et al., 1999; Cuntz et al., 2003), although the box model used in this study is much primitive compared to the 3-D models in the past studies. To investigate the possible cause(s) of the phase difference, we carried out additional simulations that incorporated three different seasonally varying $\delta_{LW}(^{18}O)$ values into the box model.

Figure 7d shows the simulated results along with the one-harmonic, best-fit curve to the observed data. It is apparent from this figure that the appearance of the seasonal minimum/maximum of $\delta_{atm}(^{18}O)$ depended on the seasonal variations of $\delta_{LW}(^{18}O)$. It is also apparent that the observed seasonal cycle of $\delta_{atm}(^{18}O)$ was well reproduced by the simulation that incorporated the $\delta_{LW}(^{18}O)$ represented by the thick dashed blue line. The scenario for the thick dashed blue line was determined based on some past studies reported seasonal variations of $\delta_{LW}(^{18}O)$ (e.g., Welp et al., 2008; Plavcová et al., 2018; Cernusak et al., 2022; Liu et al., 2023), and other two scenarios represented by two-dot chain and dotted lines were carried out as sensitivity tests to the phase difference in seasonal $\delta_{LW}(^{18}O)$ cycle. Welp et al. (2008) observed the time series of ecosystem water pools at a soybean canopy in Minnesota, USA, from 30 May to 27 September 2006 and found the most extreme enrichment of bulk $\delta_{LW}(^{18}O)$ to be 20 ‰ above xylem water during the early part of the growing season (Fig. 1a in Welp et al., 2008). Plavcová et al. (2018) and Liu et al. (2023) also reported less enrichment of $\delta_{LW}(^{18}O)$ toward the end of the vegetation season by about 10–20 ‰, and Cernusak et al. (2022) reported a strong negative correlation between the $\delta_{LW}(^{18}O)$ and the relative humidity of air based on a recent global meta-analysis. These characteristics are roughly consistent with the $\delta_{LW}(^{18}O)$ represented by the thick dashed blue line in Fig. 7d, which shows decreases of $\delta_{LW}(^{18}O)$ toward the end of the vegetation season similar in magnitude to decreases reported in the past studies. In this regard, Murayama et al. (2010) observed $\delta_{CO2}(^{18}O)$ at a forest site in Japan and reported monthly mean $\delta_{CO2}(^{18}O)$ correlated positively with $\delta(^{18}O)$ of precipitation ($\delta_{precip}(^{18}O)$). Since variations in $\delta_{LW}(^{18}O)$ are closely related to those in $\delta_{precip}(^{18}O)$, it is suggested that $\delta_{LW}(^{18}O)$ is an important driver to modify seasonal cycles both for $\delta_{atm}(^{18}O)$ and $\delta_{CO2}(^{18}O)$.

We also carried out additional box-model simulations that incorporated the average monthly $\delta_{LW}(^{18}O)$ around TKB (36° N, 140° E) and at lower latitude (30°S – 30°N) calculated by MIROC5-iso for the period 2013–2022. The results are plotted in Fig. 7d, and both the monthly $\delta_{LW}(^{18}O)$ and simulated seasonal $\delta_{atm}(^{18}O)$ cycle fall within the range of those represented by blue solid, dashed, two-dot chain, and dotted lines in the figure discussed above. Another factor to change the simulated seasonal $\delta_{atm}(^{18}O)$ cycle is the choice of the isotopic effects from B94 or L&B11 (Table 1). If we use the isotopic effects from L&B11 and ignore $R_{Res}$ and $R_{PS}$ (i.e. we consider marine respiration/production only), then the seasonal amplitude of the

simulated $\delta_{atm}(^{18}O)$ increase by 20% compared with that simulated by using the isotopic effects from B94. This is due to the difference in the isotopic effects of ocean respiration, which are 18.9 and 23.5 ‰ for B94 and L&B11, respectively. Such difference will become apparent in the southern hemisphere, where seasonal $\delta(O_2/N_2)$ cycle is driven mainly by air-sea $O_2$ flux (e.g. Keeling and Manning, 2014). Therefore, spatiotemporal variations in the seasonal $\delta_{atm}(^{18}O)$ cycle will be useful to

constrain not only spatiotemporal variations of $\delta_{LW}(^{18}O)$ but also the isotopic effects of ocean respiration.

### 3.3 Secular trend in $\delta_{atm}(^{18}O)$

Figure 8 shows temporal changes of the annual average $\delta_{atm}(^{18}O)$ observed at TKB. The error band denotes the ±0.9 per meg of the long-term stability of $\delta_{atm}(^{18}O)$ in our standard air (Fig. 2). It is apparent in Figs. 8a and 8b that the $\delta_{atm}(^{18}O)$ underwent

a slight secular increase of $(0.22 \pm 0.14)$ per meg $a^{-1}$ throughout the observation period. This rate was calculated from the difference between the 2013 and 2022 annual average $\delta_{atm}(^{18}O)$, and the uncertainty around the long-term stability was taken into account. The observed secular increasing trend was quite different from the secular decrease of the DME expected by Seibt et al. (2005), which was on the order of 70 per meg over the last 150 years ($-0.5$ per meg $a^{-1}$). They calculated the secular change by assuming anthropogenic changes of the terrestrial oxygen cycle from pre- to post-industrial times: (1) a replacement

of 3% of terrestrial respiratory $O_2$ release by biomass burning, (2) a 5% decrease of global terrestrial GPP, (3) a decrease of global photorespiration due to the increase of the amount fraction of atmospheric $CO_2$ by 100 μmol $mol^{-1}$, (4) a 10 % decrease of stomatal conductance resulting from $CO_2$ increases and a partial offset of photorespiratory decreases, and (5) a 5 % decrease of the $O_2$ flux–weighted $^{18}O$ enrichment of foliage water due to higher contributions of $^{18}O$-depleted northern mid-latitude biomes. The fact that the observation period of 10 years in the present study was much shorter than the 150 years discussed in

Seibt et al. (2005) makes it difficult to discuss the significance of the difference between the secular trends in the two studies. It would nevertheless be of interest to see if the observed secular trend could be reproduced using our box model. In that case we could explore the applicability of the precise observations of the $\delta_{atm}(^{18}O)$.

To explore that possibility, we carried out calculations in which we assumed that there were long-term changes of (1) GPP, (2) photorespiration, and (3) $\delta_{LW}(^{18}O)$. Note that we considered long-term changes of only terrestrial fluxes for the GPP and

photorespiration. Changes of marine photosynthetic/respiratory $O_2$ fluxes should be included in more detailed future studies.

We first assumed that the global terrestrial GPP increases in proportion to the global average $CO_2$ amount fraction. As the global average $CO_2$ amount fraction, we used the data from Scripps $CO_2$ Program (https://scrippsco2.ucsd.edu/data/atmospheric_co2/icecore_merged_products.html) based on ice-core data and direct observations before and after 1959, respectively (Keeling et al., 2001; Rubino et al., 2019). We assume the initial terrestrial

production of $O_2$ in 1871 as 16.7 Pmol $a^{-1}$ (Table 1), which corresponds to 107 Pg $a^{-1}$ (C equivalents) of global terrestrial GPP considering the $r_{DR}$ of 0.59 and the $OR_B$ of 1.1. Then, the GPP increased secularly with increasing $CO_2$ amount fraction, and

it takes 141 Pg a$^{-1}$ (C equivalents) in 2006. Although this is somewhat larger than the average GPP of 125 Pg a$^{-1}$ (C equivalents) for the period 1992–2020 reported by Bi et al. (2022), it falls within a range of the global GPP estimates from various models summarized in Fig. 10 of Zheng et al. (2020). A similar increase of global GPP during the 20th century has also been reported by Campbell et al. (2017) based on long-term atmospheric carbonyl sulfide (COS) records derived from ice-core, firn, and ambient air samples.

Second, we assumed an increase of 120 μmol mol$^{-1}$ in the amount fraction of atmospheric $CO_2$ during the 150 years from pre-industrial times to the present. This increase caused a decrease of global average photorespiration based on Farquhar et al. (1980):

$$\phi = \left(V_{O\_max}/V_{C\_max}\right) \times \left(y(O_2)/y(CO_2)\right) \times \left(K_C/K_O\right) \times 10^{-3} \qquad (9)$$

where $\phi$ is the ratio of photorespiration to carboxylation (or total carbon fixation, the amount of which corresponds to the sum of GPP, photorespiration, and the Mehler reaction), $(V_{O\_max}/V_{C\_max})$ is the ratio of the maximum oxygenation velocity to the maximum carboxylation velocity of RuP$_2$ carboxylase-oxygenase (we used 0.21 for this ratio from equation 16 in Farquhar et al. (1980)). $y(O_2)$ and $y(CO_2)$ are amount factions of $O_2$ and $CO_2$, respectively, in equilibrium with their dissolved amount fractions in the chloroplast stroma. We used atmospheric $y(O_2)$, and atmospheric $y(CO_2)$ multiplied by 0.7 following B94. $(K_C/K_O)$ is the ratio of the Michaelis-Menten constants for carboxylation and oxygenation, respectively (we adopted 460/330 for this ratio from Table 1 in Farquhar et al. (1980)). $10^{-3}$ is a coefficient to compare the calculated $\phi$ with those in Farquhar et al. (1980) directly since they used units of mbar and μbar for the $y(O_2)$ and $y(CO_2)$, respectively. We calculated $\phi$ to be 0.31 and 0.22 for $CO_2$ amount fractions of 280 and 400 μmol mol$^{-1}$, respectively.

We then calculated changes of $\delta_{LW}(^{18}O)$ with MIROC5-iso for the period 1871–2022. We considered the water cycle to be in steady state before 1871, and we assumed the global average $\delta_{LW}(^{18}O)$ in 1871 to be 4.4 or 6.5 ‰ based on previous studies for the DME in steady state by B94 or L&B11, respectively. In this connection, Hoffmann et al. (2004) have reported an intermediate $\delta_{LW}(^{18}O)$ of 5–6 ‰. It should be noted that the original $\delta_{LW}(^{18}O)$ calculated by MIROC5-iso for 1871 was about –0.7 ‰, so that we arbitrarily shifted all $\delta_{LW}(^{18}O)$ calculated by MIROC5-iso by 5.1 or 7.2 ‰. Clarifying the cause(s) of the low $\delta_{LW}(^{18}O)$ calculated by MIROC5-iso will be a future task. Figure 8c shows the global average $\delta_{LW}(^{18}O)$; $\delta_{LW}(^{18}O)$ underwent a significant secular increase throughout the period. The increase was especially clear after the 1980s, which is the time when there was an increase of the $\delta_{precip}(^{18}O)$ simulated by MIROC5-iso (not shown). Rozanski et al. (1992) have reported that $\delta_{precip}(^{18}O)$ increases with increasing surface air temperature by 0.6 ‰ K$^{-1}$, and the global average surface air temperature has increased by about 1 K from 1980 to the present. The increase of surface air temperature therefore caused at least part of the simulated secular increase of $\delta_{LW}(^{18}O)$ since 1980. Previous studies have also reported that a lower (larger) relative humidity near the plant stomata enhances (diminishes) $\delta_{LW}(^{18}O)$ (eq. (2) of Hoffmann et al. (2004)). Also, Byrne and

O'Gorman (2018) have reported that the relative humidity over land from 40°S to 40°N has decreased secularly since 1980. It is therefore possible that the decrease of relative humidity also contributed to the simulated secular increase of $\delta_{LW}(^{18}O)$ since 1980. It should be noted that Welp et al. (2011) suggested that $\delta_{CO2}(^{18}O)$ increases with increasing $\delta_{precip}(^{18}O)$ and $\delta_{LW}(^{18}O)$ through the redistribution of moisture and rainfall in the tropics during an El Niño, which leads to substantial interannual variations in $\delta_{CO2}(^{18}O)$ during 1977–2009 obtained from the Scripps Institution of Oceanography global flask network. Therefore, it will be important in future studies to examine not only secular trend discussed in this study but also interannual variations in $\delta_{LW}(^{18}O)$ and $\delta_{atm}(^{18}O)$.

Figures 8a and 8b show $\delta_{atm}(^{18}O)$ simulated by the box model that incorporated the above-mentioned, long-term changes of GPP, photorespiration, and $\delta_{LW}(^{18}O)$. The observed and simulated secular trends in $\delta_{atm}(^{18}O)$ agreed well with each other, under the conditions assuming the isotopic effects from B94. On the other hand, the simulated $\delta_{atm}(^{18}O)$ assuming the isotopic effects from L&B11 decreased secularly, contrary to the observed secular increase. Figure 8b shows the respective contributions of the changes of GPP, photorespiration, and $\delta_{LW}(^{18}O)$ to the simulated $\delta_{atm}(^{18}O)$. The simulated $\delta_{atm}(^{18}O)$ showed secular increases with increasing GPP during 1871–2023, and the increase is much larger in the simulation assuming the isotopic effects from B94 than that from L&B11. The simulated $\delta_{atm}(^{18}O)$ showed secular increases with increasing $\delta_{LW}(^{18}O)$ for both the cases assuming isotopic effects from B94 and L&B11. This pattern differed from the results simulated by Seibt et al. (2005), who reported a secular decrease of $\delta_{atm}(^{18}O)$ based on assumed secular decreases of GPP and $\delta_{LW}(^{18}O)$ during the last 150 years. It is noted that the contributions of the changes of $\delta_{LW}(^{18}O)$ to the simulated $\delta_{atm}(^{18}O)$ increased with time monotonously while clear increase of $\delta_{LW}(^{18}O)$ was found after the 1980s (Figs. 8b–c). This is due to the choice of the initial $\delta_{LW}(^{18}O)$ in 1871; we set it to be 4.4 or 6.5 ‰ (the values for steady state by B94 or L&B11). As seen from Fig. 8c, the average $\delta_{LW}(^{18}O)$ during 1872–1980 was higher than the initial values, which made the monotonous increase of the $\delta_{atm}(^{18}O)$ driven by the $\delta_{LW}(^{18}O)$ changes. In contrast, both the present study and that of Seibt et al. (2005) found a secular decrease of the simulated $\delta_{atm}(^{18}O)$ with decreasing ratio of photorespiration to carboxylation ($\phi$).

The contributions of $\phi$ and $\delta_{LW}(^{18}O)$ almost cancelled each other in the simulation assuming the isotopic effects from B94. As a result, the simulated $\delta_{atm}(^{18}O)$ based on B94 increased secularly due mainly to the contribution of the secular increase of GPP. On the other hand, the contribution of GPP to the simulated $\delta_{atm}(^{18}O)$ assuming the isotopic effects from L&B11 is much smaller. Moreover, the secular decrease of the simulated $\delta_{atm}(^{18}O)$ due to the contribution of $\phi$ is larger for the simulation assuming the isotopic effects from L&B11 than that from B94. As a result, the simulated $\delta_{atm}(^{18}O)$ based on L&B11 decreased secularly due mainly to the contribution of the secular decrease of $\phi$. The substantial difference between the contributions of GPP for the simulations based on B94 and L&B11 are attributed to the differences in the terrestrial/oceanic DME in their studies. Specifically, the respective terrestrial and oceanic DME in steady states are 22.4 and 18.9 ‰ in B94, while they are 23.5 and 23.5 ‰ in L&B11 (Table 1). Therefore, the secular increase of terrestrial production incorporated into our simulations

lead to the secular increase of $\delta_{atm}(^{18}O)$ for the isotopic effects based on B94. The substantial difference in the contributions of $\phi$, found between the simulations based on B94 and L&B11, is attributed to the increase of $r_{DR}$ accompanied with decrease of $\phi$ and the larger isotopic effect for dark respiration in B94 (18 ‰) than that in L&B11 (15.8 ‰). Therefore, we confirmed that secular trends in the simulated $\delta_{atm}(^{18}O)$ are highly sensitive to the isotopic effects associated with the DME. This means that further studies are needed to determine the isotopic effects precisely, in order to evaluate long-term changes in GPP and photorespiration based on $\delta_{atm}(^{18}O)$. In this regard, some past studies evaluated the mechanisms responsible for the increase of global GPP. For example, Madani et al. (2020) have reported an increase of GPP in northern latitudes caused by a reduction of cold-temperature constraints on plant growth. This scenario suggests that there has been an increase of negative carbon-climate feedback in high latitudes, whereas there has been a suggestion of an emerging positive climate feedback in the tropics, mainly due to an increase of the atmospheric vapor pressure deficit. They have also pointed out that models have been struggling to determine how much additional $CO_2$ is being taken up by plants as a result of increased amount fractions of atmospheric $CO_2$. Therefore, an analysis based on a secular change in $\delta_{atm}(^{18}O)$, which enables estimation of changes of the ratios of carboxylation to global GPP and photorespiration to global GPP, will facilitate better understanding of global $CO_2$ fertilization processes.

We used the global average secular change of $\delta_{LW}(^{18}O)$ simulated by the MIROC5-iso in this analysis, and we found that it made a substantial contribution to the simulated $\delta_{atm}(^{18}O)$. The implication is that the secular change of the water cycle must be accurate before the observed and simulated secular trends of $\delta_{atm}(^{18}O)$ can be equated. In other words, $\delta_{atm}(^{18}O)$ is a unique tracer for a comprehensive evaluation of global changes of the oxygen, carbon, and water cycles. For example, if the secular increase of the global average amount fraction of atmospheric $CO_2$ stops without changes of the secular increasing trends of GPP, then the global average $\delta_{atm}(^{18}O)$ will increase faster than the rate shown in Fig. 8a by assuming the isotopic effects from B94. Substantial secular decrease of the global average $\delta_{atm}(^{18}O)$ may also be expected under pessimistic scenarios, such as substantial deforestation (secular decrease of GPP) and an increase of the average global amount fraction of atmospheric $CO_2$. In both cases, the results are regulated by climate changes such as changes of surface air temperature and aridification that lead to secular changes of $\delta_{LW}(^{18}O)$.

We recognize that secular changes of stratospheric gravitational separation may cause slight secular changes of the surface $\delta_{atm}(^{18}O)$. Ishidoya et al. (2021) have estimated this effect for atmospheric $\delta(Ar/N_2)$ at the surface to be –0.13 and 0.15 per meg $a^{-1}$ when accompanied by a weakening or enhancement of the Brewer–Dobson circulation, respectively. These values correspond to –0.02 and 0.03 per meg $a^{-1}$, respectively, for $\delta_{atm}(^{18}O)$ if mass-dependent gravitational separation is assumed. The secular trend of $\delta_{atm}(^{18}O)$ due to changes of stratospheric gravitational separation is negligible at present because the changes are much smaller than the uncertainty of the secular trend of the observed $\delta_{atm}(^{18}O)$ at TKB ((0.22 ± 0.14) per meg $a^{-}$

[1]). If the observation period increases, the uncertainty of the secular trend will be smaller. Consideration of stratospheric gravitational separation changes may therefore be needed in future.

## 4 Conclusions

We have carried out high-precision measurements of $\delta_{atm}(^{18}O)$ at TKB site since 2013. Clear variations of $\delta_{atm}(^{18}O)$ with a daytime minimum were found for the average diurnal cycles throughout the observation period. The much larger amplitudes of the diurnal $\delta_{atm}(^{18}O)$ cycles in summer than in winter suggest a substantial contribution of the activities in the terrestrial biosphere to the diurnal cycle. The amplitudes and phases of the diurnal $\delta_{atm}(^{18}O)$ and $\delta(O_2/N_2)$ cycles simulated by a box model, which incorporated the terrestrial oxygen cycle, were roughly consistent with the observed diurnal cycles in summer. Seasonal changes of the ERs, calculated from the average diurnal cycles of the $\delta(O_2/N_2)$ and amount fractions of $CO_2$, also indicated a larger contribution of the activities in the terrestrial biosphere in summer than in winter. We found that the 5h- and 19h-averaged $\delta_{atm}(^{18}O)$ also varied in antiphase with $\delta(O_2/N_2)$ in summer. We found that the diurnal cycles of $\delta_{BIO}(O_2/N_2)$ and $\delta_{FF}(O_2/N_2)$, estimated by the $\delta_{atm}(^{18}O)$-method, agreed well with the diurnal $\delta(O_2/N_2)$ cycles driven by activities in the terrestrial biosphere and fossil fuel combustion estimated by the ER-method.

The $\delta_{atm}(^{18}O)$ varied seasonally in antiphase with $\delta(O_2/N_2)$ and was a minimum in the summer. We found the peak-to-peak amplitude of the average seasonal $\delta_{atm}(^{18}O)$ cycle to be about 2 per meg. These characteristics were generally reproduced by the box model, and the seasonal $\delta_{atm}(^{18}O)$ cycle was driven mainly by an input of photosynthetic $O_2$, the $\delta(^{18}O)$ of which was about 20 ‰ lower than $\delta_{atm}(^{18}O)$. The box model also suggested that the seasonal cycle of $\delta_{atm}(^{18}O)$ was substantially affected by seasonally varying $\delta_{LW}(^{18}O)$, which indicated the usefulness of $\delta_{atm}(^{18}O)$ observations to constrain spatiotemporal variations of $\delta_{LW}(^{18}O)$. There was a secular increase of the $\delta_{atm}(^{18}O)$ by $(0.22 \pm 0.14)$ per meg $a^{-1}$ throughout the observation period. To interpret the secular trend, we used the box model that incorporated two kinds of the isotopic effects associated with the DME from B94 and L&B11, to carry out simulations in which we considered the long-term changes of GPP, photorespiration, and $\delta_{LW}(^{18}O)$. For the calculation of $\delta_{LW}(^{18}O)$, we also used the 3-D model MIROC5-iso. We found that all three components made substantial contributions to the simulated $\delta_{atm}(^{18}O)$. If we assume the isotopic effects from B94, then the observed secular increase of $\delta_{atm}(^{18}O)$ was reproduced by the simulation mainly due to the secular increase of GPP. On the other hand, the simulated $\delta_{atm}(^{18}O)$ based on L&B11 decreased secularly due mainly to the contribution of the secular decrease of $\phi$. The substantial difference between the secular $\delta_{atm}(^{18}O)$ changes in the simulations based on B94 and L&B11 are attributed not only to the differences in the terrestrial/oceanic DME but also to the larger isotopic effect for dark respiration in B94. Therefore, further studies are needed to determine the isotopic effects for the DME precisely.

In conclusion, we confirmed that precise observations of the spatiotemporal variations of $\delta_{atm}(^{18}O)$ will enable better understanding of the global cycles of $O_2$, $CO_2$, and water. However, no relevant observational results have previously been reported. Additional steps should therefore include observations of the $\delta_{atm}(^{18}O)$ at some surface stations in both hemispheres using continuous measurement systems that are similar to the system used in the present study and a newly developed, precise measurement system for flask samples. Two- and three-dimensional models to calculate $\delta_{atm}(^{18}O)$ should be developed to interpret the latitudinal differences of the observed $\delta_{atm}(^{18}O)$ variations. There is also need for improvement of the three-dimensional model simulation of $\delta_{LW}(^{18}O)$, because the original $\delta_{LW}(^{18}O)$ calculated with MIROC5-iso was systematically lower than those reported by past studies. Such progress will better enable detection of the signal of climate changes associated with the DME.

## Appendix A: A list of symbols

There are a lot of symbols used in the present study especially for the model simulations. For readers' convenience, names, definitions, and units of the symbols are summarized in Table A1.

## Appendix B: Mass balance equation for DME

In the following formulas, capital letters represent values related to $^{16}O^{16}O$ and lowercase letters represent values related to $^{18}O^{16}O$, respectively. Here we define the total amounts of atmospheric $^{16}O^{16}O$ and $^{18}O^{16}O$ as $M$ and $m$, respectively. The DME is a result caused by mixed processes of the sinks and sources. The sinks oxidize by using atmospheric $O_2$, which usually cause isotopic fractionation. On the other hand, the source adds $O_2$ to the atmosphere with an isotopic ratio that is usually independent of atmospheric $O_2$. Here we define the fluxes of source are $F_{Si}$ and $f_{Si}$, and the fluxes of sink processes are $F_{Lj}$ and $f_{Lj}$, respectively. (i and j represent the types of sources and sinks, respectively.) The source isotopic ratio, $\Delta_i$ is expressed by the following equation.

$$\Delta_i = \frac{f_{Si}/F_{Si}}{r_{std}} - 1 \quad (A1).$$

Here, $r_{std}$ is the $^{18}O^{16}O/^{16}O^{16}O$ ratio of the standard material. Sink fluxes can be expressed simply by using reaction rates $K_j$ and $k_j$ as a first order approximation:

$$F_{Lj} = -K_j y(^{16}O^{16}O), \quad f_{Lj} = -k_j y(^{18}O^{16}O) \quad (A2).$$

Here, $y(^{16}O^{16}O)$ and $y(^{18}O^{16}O)$ represent amount fractions of atmospheric $^{16}O^{16}O$ and $^{18}O^{16}O$, respectively. For the one-box model, the ratio of amount fractions is equal to that of the total amounts as follows,

$$\frac{m}{M} = \frac{y(^{18}O^{16}O)}{y(^{16}O^{16}O)} \quad (A3).$$

The isotopic ratio of atmospheric $O_2$ is defined as follows:

$$\delta = \frac{1}{r_{std}} \frac{y(^{18}O^{16}O)}{y(^{16}O^{16}O)} - 1 \quad (A4).$$

The enrichment factor, $\varepsilon_j$ is defined as follows:

$$\varepsilon_j = 1 - \frac{k_j}{K_j} \quad (A5).$$

The time derivative formula of the isotopic ratio of the atmospheric $O_2$, $\delta$ is

630 $\quad \frac{d\delta}{dt} = \frac{1}{r_{std}M}\left(\frac{dm}{dt} - \frac{m}{M}\frac{dM}{dt}\right) \quad (A6).$

Here, the mass balance equations for $^{16}O^{16}O$ and $^{18}O^{16}O$ are

$$\frac{dM}{dt} = \sum_i F_{Si} + \sum_j F_{Lj} \quad \text{and} \quad \frac{dm}{dt} = \sum_i f_{Si} + \sum_j f_{Lj} \quad (A7)$$

, respectively. Substituting these into equation (A6) and rearranging using the source isotopic ratios, $\Delta_i$ and the enrichment factor $\varepsilon_j$, we get

$\quad \frac{d\delta}{dt} = \sum_i \frac{F_{Si}}{M}(\Delta_i - \delta) - \sum_j \left\{\frac{F_{Lj}}{M}(\delta + 1)\varepsilon_j\right\} \quad (A8).$

It can be approximated as follows:

$$(\delta + 1)\varepsilon_j \cong \varepsilon_j \quad (A9),$$

and we define the ratios of $F_{Si}/M$ and $F_{Lj}/M$ as $R_{Si}$ and $R_{Lj}$, respectively, then we get

$$\frac{d\delta}{dt} = \sum_i R_{Si}(\Delta_i - \delta) - \sum_j R_{Lj}\varepsilon_j \quad (A10).$$

Equation (3) is obtained by applying equation (A10) to the various sources/sinks. Note that the sink ratio, $R_{Lj}$ is a negative value. Since $\varepsilon_j > 0$, $-R_{Lj} \times \varepsilon_j$ becomes positive value, which results in enrichment for the atmosphere. As an exception, $\varepsilon_{ST}$ is given as the isotope effect during transport from the stratosphere to the troposphere, and $\varepsilon_{ST} < 0$. Conversely, the same stratospheric effect would be achieved if $\varepsilon_{TS}$ was given as a positive value and $\varepsilon_{ST}$ was set to zero.

The $\delta$ value in a steady state ($\delta_{ss}$) can be obtained from the above formula as follows:

$\quad \delta_{ss} = \frac{\sum_i R_{Si}\Delta_i - \sum_j R_{Lj}\varepsilon_j}{\sum_i R_{Si}} \quad (A11).$

For example, if we simply consider only photosynthesis and respiration in terrestrial and oceanic biosphere, and assume that $R_{PS} = -R_{RES}$ and $R_{OP} = -R_{OR}$, respectively (here we use the same variables in Table 1), equation (A10) becomes the following equation.

$$\frac{d\delta}{dt} = R_{PS}(\Delta_{PS} + \varepsilon_{RES} - \delta) + R_{OP}(\Delta_{OW} + \varepsilon_{OR} - \delta) \quad (A12)$$

In a steady state, (A12) becomes as follows:

$$\delta_{ss} = \frac{R_{PS}(\Delta_{PS} + \varepsilon_{RES}) + R_{OP}(\Delta_{OW} + \varepsilon_{OR})}{R_{PS} + R_{OP}} \quad (A13).$$

If we substitute the relative ratios of the terrestrial and oceanic fluxes, $\lambda_{ter} = R_{PS} / (R_{PS} + R_{OP})$ and $\lambda_{oc} = R_{OP} / (R_{PS} + R_{OP})$, respectively into (A13), we get

$$\delta_{ss} = \lambda_{ter}(\Delta_{PS} + \varepsilon_{RES}) + \lambda_{OC}(\Delta_{OW} + \varepsilon_{OR}) \quad (A14).$$

This formula is often used for simple estimation of DME to separate the terrestrial and oceanic effects.

## Data availability

The observational data for diurnal cycles and monthly mean values shown in Fig. 4, 6, and 7 are included as electronic supplement to the manuscript. We will deposit the data in an appropriate data archive before the manuscript is accepted for publication.

## Author contributions

SI designed the study, conducted measurements of $\delta_{atm}(^{18}O)$, $\delta(O_2/N_2)$ and $CO_2$ amount fractions, and drafted the manuscript. SS conducted the box model simulations. AO conducted the MIROC5-iso simulations. All authors approved the final manuscript.

## Competing interests

The authors declare that they have no conflict of interest.

## Acknowledgements

This study was partly supported by Japan Society for the Promotion of Science KAKENHI grants (grant nos. 22H05006, 23H00513, and 22H04938) and the Global Environment Research Coordination System from the Ministry of the Environment, Japan (grant no. METI1953).

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

**Table 1: Budgets and isotopic effects of the atmospheric O₂ used in the box model. Isotopic values of the external sources and DME are described versus VSMOW. B94 and L&B11 denote Bender et al. (1994) and Luz and Barkan (2011), respectively.**

| | Budget (Pmol a$^{-1}$) | Symbol | Fraction | Type of isotopic effect | Isotopic effect (permil) B94 | Isotopic effect (permil) L&B11 | Footnote |
|---|---|---|---|---|---|---|---|
| **Terrestrial production** | 16.7 | $R_{PS}$ | 4.5 x 10$^{-4}$ | | | | a |
| **Photosynthesis** | | $\Delta_{PS}$ | | External source (Leaf water) | 4.4 | 6.5 | |
| **Leaf water enrichment** | | $\varepsilon_{LE}$ | | Solubility equilibration in leaf water | 0.70 | 0.75 | |
| **Terrestrial respiration** | –16.7 | $R_{RES}$ | –4.5 x 10$^{-4}$ | | | | a |
| **Dark respiration** | | $\varepsilon_{DR}$ | ($r_{DR}$ 0.59) | Fractionation | 18 | 15.8 | b |
| **Photorespiration** | | $\varepsilon_{PR}$ | ($r_{PR}$ 0.31) | Fractionation | 21.2 | 22.0 | b |
| **Mehler reaction** | | $\varepsilon_{MR}$ | ($r_{MR}$ 0.10) | Fractionation | 15.1 | 15.1 | b |
| **Total respiration effect** | | $\varepsilon_{RES}$ | ($r_{DR}$+ $r_{PR}$+ $r_{MR}$=1.0) | Fractionation | 18.7 | 17.7 | c |
| **Oceanic production** | 9.8 | $R_{OP}$ | 2.6 x 10$^{-4}$ | | | | d |
| **Photosynthesis** | | $\Delta_{OW}$ | | External source | 0 | 0 | e |
| **Oceanic respiration** | –9.8 | $R_{OR}$ | –2.6 x 10$^{-4}$ | | | | |
| | | $\varepsilon_{OR}$ | | Fractionation | 18.9 | 23.5 | e |
| **ST exchange** | ±3.0 x 10$^{3}$ | $R_{ST}$, $R_{TS}$ | ±8.1 x 10$^{-2}$ | | | | f |
| **Strato. to tropo.** | | $\varepsilon_{ST}$ | | Fractionation | –0.0025 | –0.0025 | g |

| | | | | | | |
|---|---|---|---|---|---|---|
| **Tropo. to strato.** | | $\varepsilon_{TS}$ | Fractionation | 0 | 0 | g |
| **Total DME (Steady state)** | 0 | | | 20.82 | 23.16 | |
| **Fossil fuel combustion** | Time dependent | $R_{FF}$ | | | | |
| | | $\varepsilon_{FF}$ | Fractionation | | 0 | h |

a: 16.7 Pmol a$^{-1}$ is the value by Hoffmann et al. (2004). *R* values were calculated by assuming that total amount of atmospheric O$_2$ is 3.706 x 10$^4$ Pmol.

b: The relative ratios of dark respiration, photorespiration, and Mehler reaction, and the isotopic effect of the Mehler reaction by B94.

c: 17.7 = 0.59 x 15.8 + 0.31 x 22.0 + 0.10 x 15.1, which becomes the same value by L&B11.

d: Calculated by using the ratio of 0.63:0.37 for the fraction of O$_2$ production by L&B11.

e: Oceanic DME effect by L&B11. They showed that the total oceanic DME is 23.5 ‰ and that the fractionations of oceanic photosynthesis exist. We assume that the same fractionation occurs only by respiration to realize the same extent of oceanic DME.

f: Calculated by using the net mass flux of S-T exchange in Olsen et al (2004).

g: Stratospheric diminution effect was calculated as the $^{18}$O discrimination in stratosphere-troposphere exchange. Because the STE flux is about 100 times larger than the total of terrestrial and oceanic flux, this contribution to the DME becomes about –0.3 ‰ (–0.3 = $R_{ST}$ x $\varepsilon_{ST}$ / ($R_{PS}+R_{OP}$)), which is the same value by L&B11.

h: We assume that atmospheric oxygen is consumed without isotope effects in fossil fuel combustion considering high temperature during the industrial combustion processes.

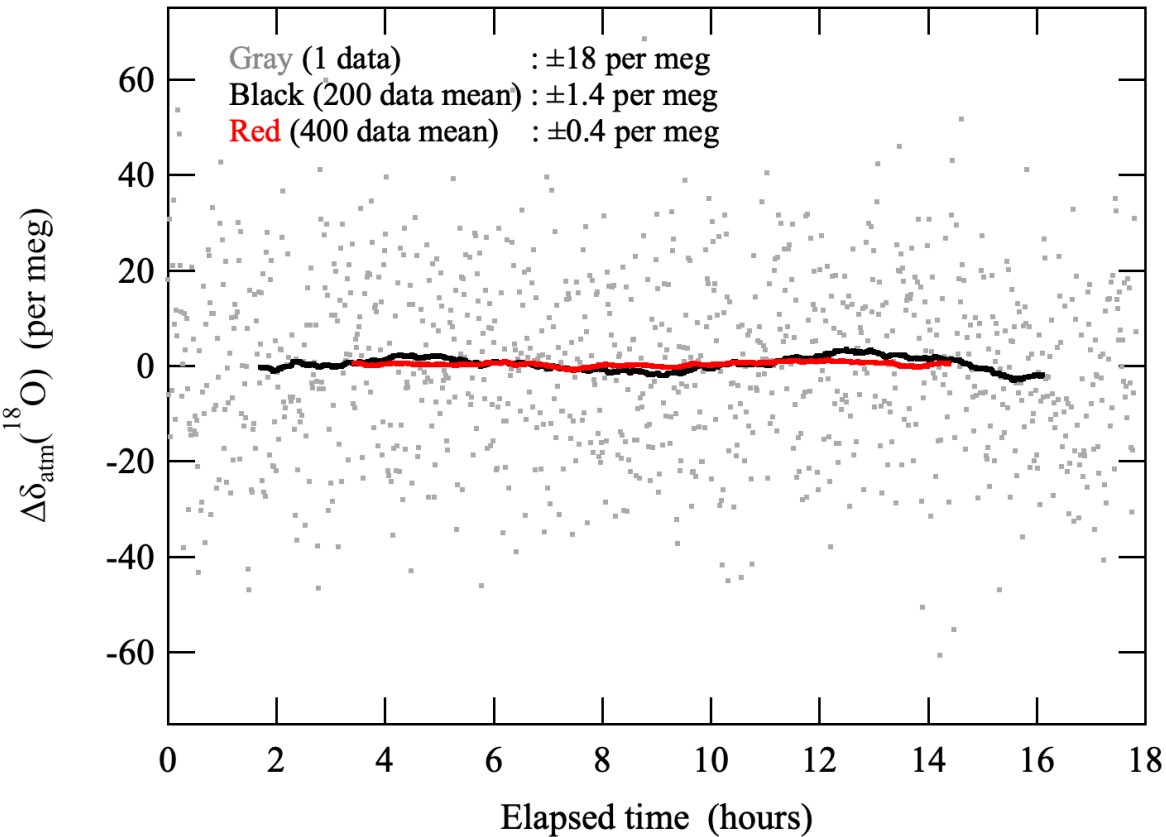

**Figure 1: Typical analytical results of the difference (Δ) of the $\delta_{atm}(^{18}O)$ of standard air against a reference air. Data are shown as deviations from the average value throughout the analysis. Gray dots, black lines, and red lines denote raw data, and averages of 200 and 400 data (corresponding to about 62 seconds, 3.5 hours, and 7 hours), respectively.**

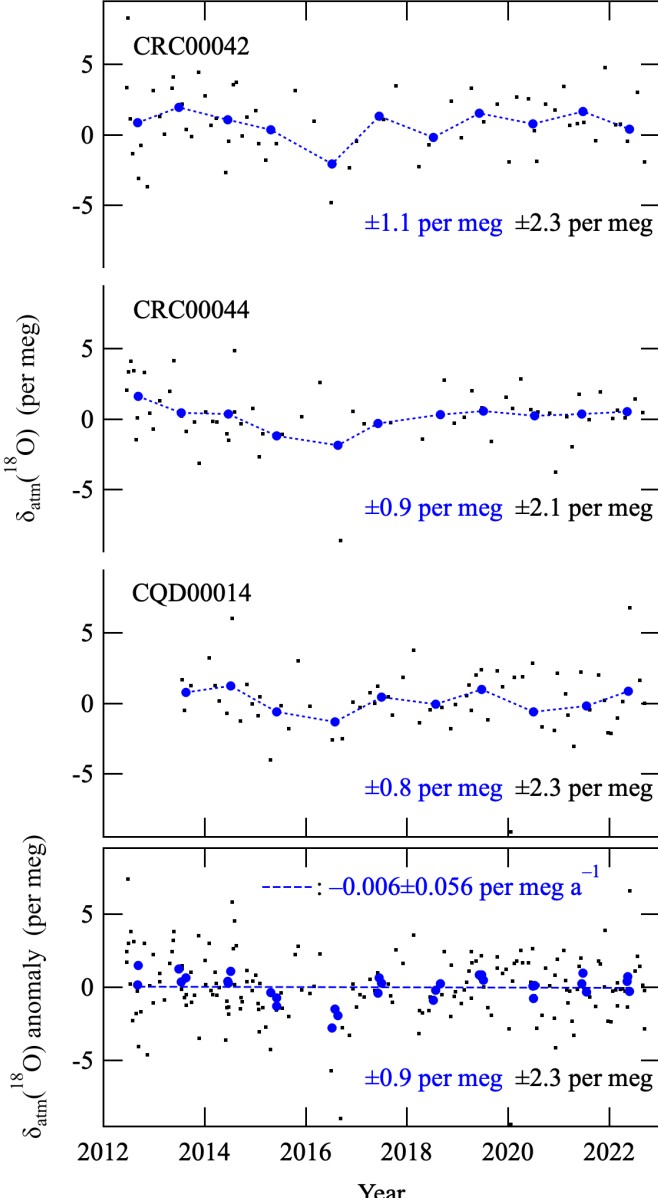

**Figure 2: (top) Each value (black dots) and the corresponding annual average (blue circles) of $\delta_{atm}(^{18}O)$ of three secondary standards against the primary standard air. (bottom) Anomalies of $\delta_{atm}(^{18}O)$ of the three secondary standards. Blue dashed line denotes the regression line fit to the data.**

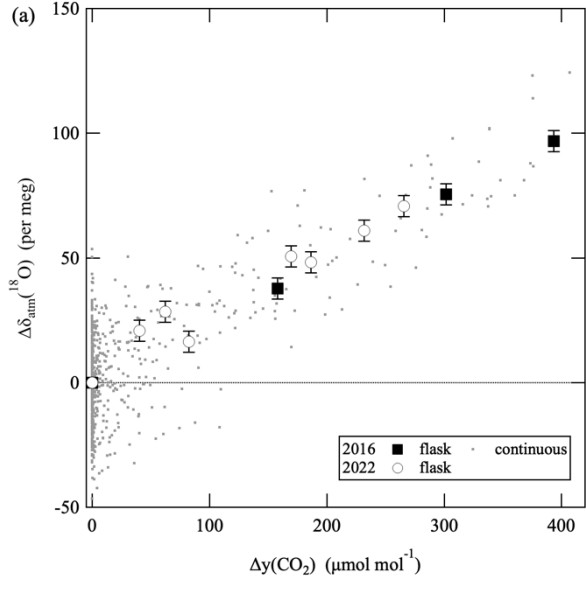

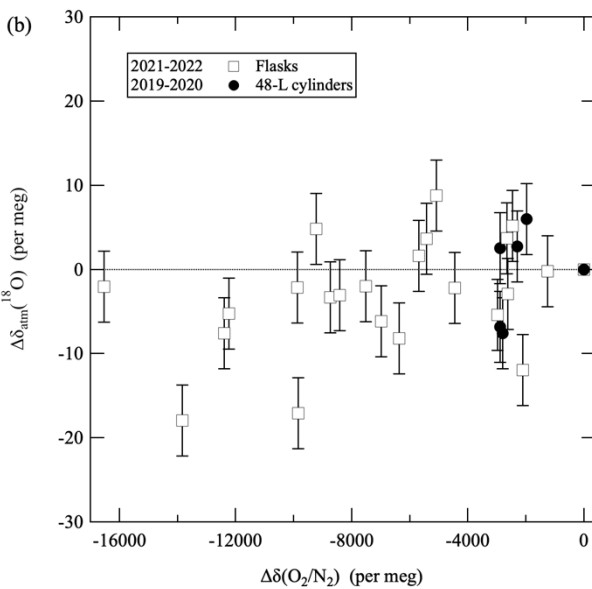

**Figure 3: (a)** Changes (Δ) of the measured $\delta_{atm}(^{18}O)$ of air samples as a function of the amount fraction of $CO_2$. $\Delta y(CO_2)$ represents the difference between the amount fraction of $CO_2$ of the air sample after and before adding pure $CO_2$. $y$ stands for the dry amount fraction of gas. **(b)** Changes of the measured $\delta_{atm}(^{18}O)$ of the air sample as a function of its $\delta(O_2/N_2)$. $\Delta\delta(O_2/N_2)$ represents the

difference between the $\delta(O_2/N_2)$ of the air sample after and before adding pure $N_2$. Error bars in (a) and (b) indicate uncertainties ($\pm 1\sigma$) for the measurements of air samples in flasks.

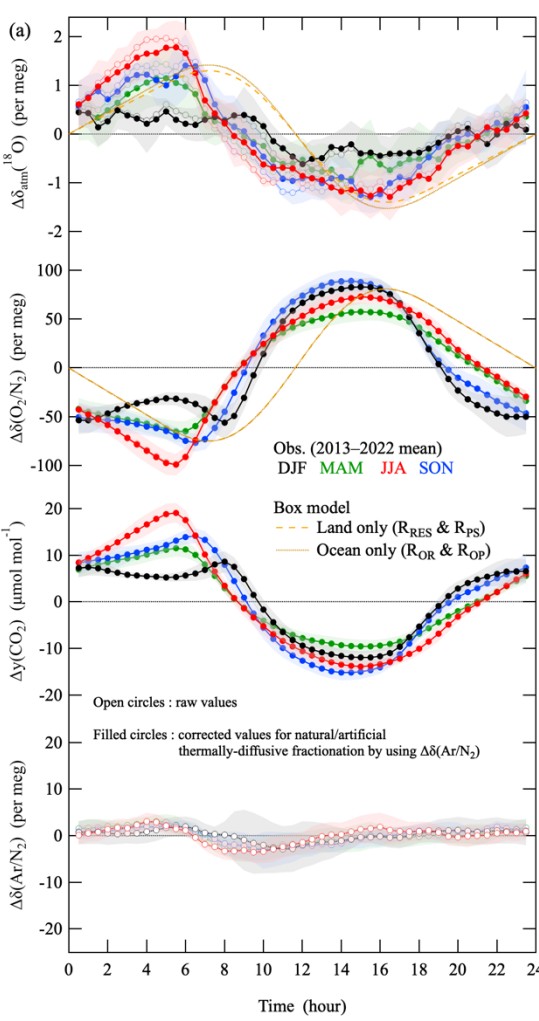

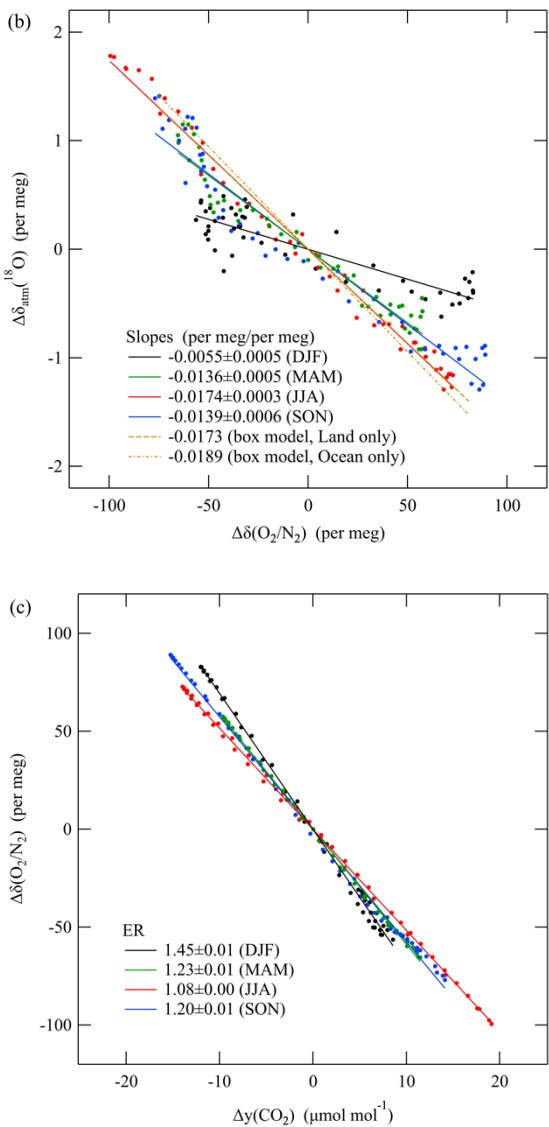

**Figure 4: (a) Plots of average diurnal cycles of $\Delta\delta_{atm}(^{18}O)$, $\Delta\delta(O_2/N_2)$, $\Delta y(CO_2)$, and $\Delta\delta(Ar/N_2)$ (open circles) observed at the TKB site during 2013–2022 for each season: December to February (black), March to May (green), June to August (red), and September to November (blue). Error bands indicate year-to-year variations during the observation periods ($\pm 1\sigma$). Those of $\Delta\delta_{atm}(^{18}O)$,**

**$\Delta\delta(O_2/N_2)$, and $\Delta y(CO_2)$, corrected for thermally diffusive fractionation, are also plotted (filled circles) (see text). The range of the vertical axis for $\Delta\delta(Ar/N_2)$ was adjusted arbitrarily to facilitate visual assessment of the variations of the observed $\Delta\delta_{atm}(^{18}O)$ due to thermally diffusive fractionation. Average diurnal cycles of $\Delta\delta_{atm}(^{18}O)$ and $\Delta\delta(O_2/N_2)$ that were simulated with a box model are also shown. The simulated values that considered only terrestrial and marine respiration/production are shown by solid and dashed ocher lines, respectively (see text). $\Delta$ denotes deviations from the diurnal mean values. (b) Relationships between $\Delta\delta_{atm}(^{18}O)$ and**

$\Delta\delta(O_2/N_2)$ for the data corrected for thermally diffusive fractionation in (a). Regression lines fitted to the observed and simulated data are also shown. (c) Same as in (b) but for the relationship between $\Delta\delta(O_2/N_2)$ and $\Delta y(CO_2)$.

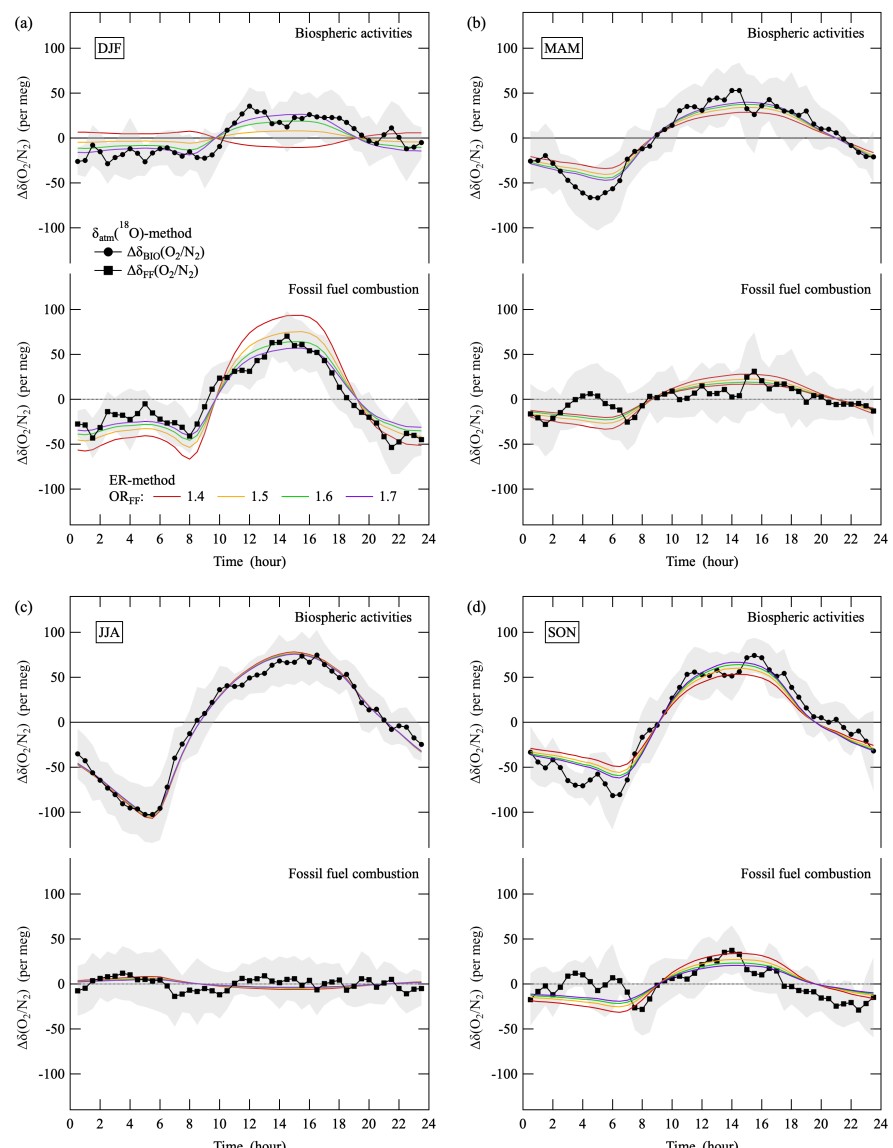

**Figure 5: (a) Plots of average diurnal cycles of the $\Delta\delta_{BIO}(O_2/N_2)$ and $\Delta\delta_{FF}(O_2/N_2)$ for each season estimated by the $\delta_{atm}(^{18}O)$-method. $\Delta\delta(O_2/N_2)$ driven by activities in the terrestrial biosphere and fossil fuel combustion estimated by the ER-method are also shown. See text for details of the $\delta_{atm}(^{18}O)$- and ER-methods. $\Delta$ denotes deviations from the diurnal mean values. Error bands for $\Delta\delta_{BIO}(O_2/N_2)$ are derived from $\Delta\delta_{atm}(^{18}O)$ in Figure 4a. Error bands for $\Delta\delta_{FF}(O_2/N_2)$ are assumed to be the same as those for**
**$\Delta\delta_{BIO}(O_2/N_2)$.**

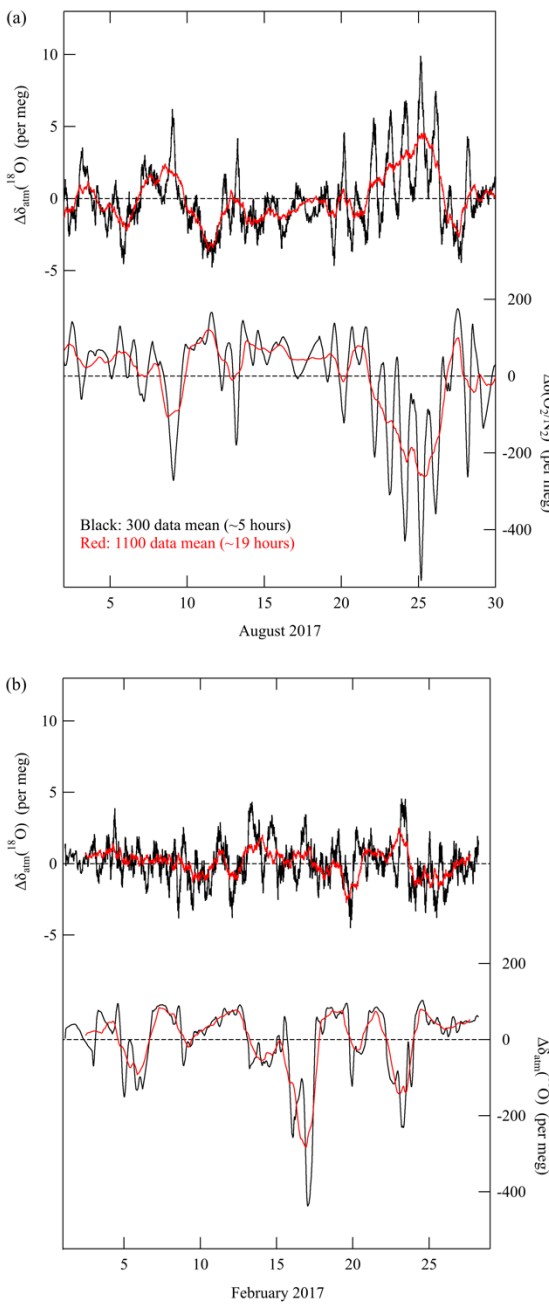

**Figure 6: (a) Rolling average values of $\Delta\delta_{atm}(^{18}O)$ and $\Delta\delta(O_2/N_2)$ for 300 data (black) and 1100 data (red) at TKB in August 2017. $\Delta$ denotes deviations from the monthly mean value. (b) Same as in (a), but for February 2017.**

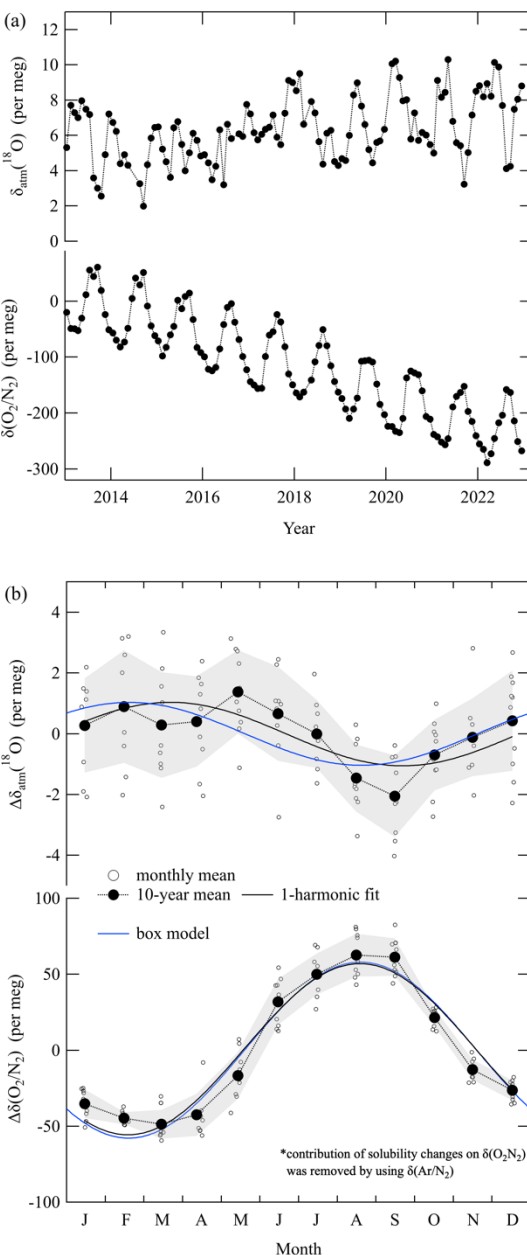

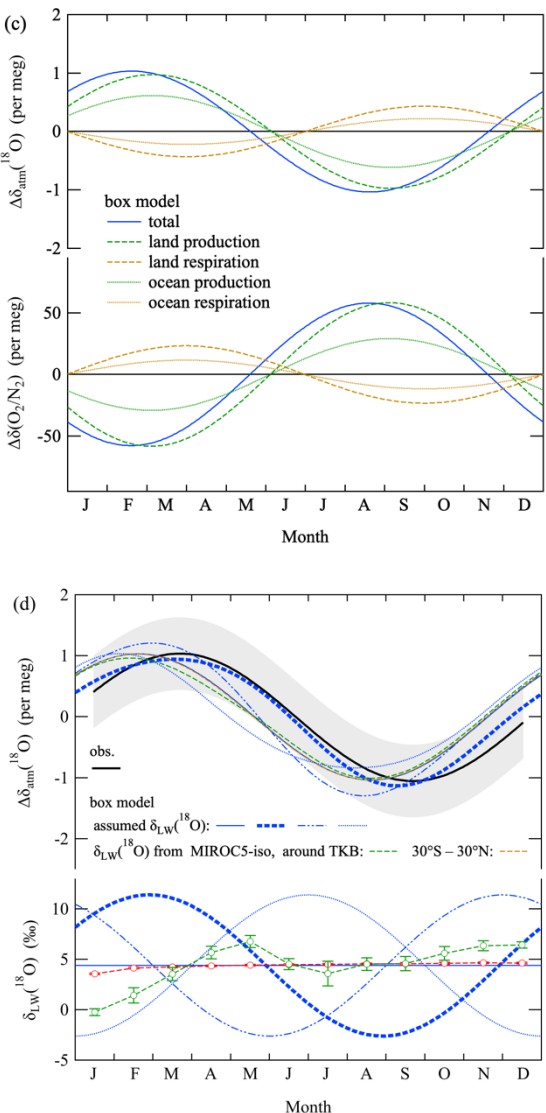

Figure 7: **(a) Monthly mean values of $\delta_{atm}(^{18}O)$ and $\delta(O_2/N_2)$ observed at TKB for the period 2013–2022. Local effects of fossil fuel combustion around TKB were excluded from $\delta(O_2/N_2)$. See text for details. (b) Detrended monthly mean values (open black circles) and their 10-year average (filled black circles) of $\Delta\delta_{atm}(^{18}O)$ at TKB for the period 2013–2022. Those of $\Delta\delta(O_2/N_2)$, extracted by**

965 **removing the contributions of solubility change by using the average seasonal $\delta(Ar/N_2)$ cycle at TKB (see text), are also shown. Average seasonal cycles of $\Delta\delta_{atm}(^{18}O)$ and $\Delta\delta(O_2/N_2)$ obtained by applying one-harmonic best-fit curves to the data (solid black lines) and those simulated by the box model (solid blue lines) are also shown. We assumed a constant $\delta_{LW}(^{18}O)$ of 4.4 ‰ for the simulation. $\Delta$ denotes deviations from the annual mean values. (c) Same average seasonal cycles of $\Delta\delta_{atm}(^{18}O)$ and $\Delta\delta(O_2/N_2)$ simulated by the box model in (b) (blue solid lines), and the respective contributions of terrestrial production (green dashed lines), terrestrial**

**respiration (brown dashed lines), ocean production (green dotted lines), and ocean respiration (brown dotted lines). (d) Same best-fit curve for $\delta_{atm}(^{18}O)$ as in (b). Error bands indicate average deviations from the 10-year average of $\Delta\delta_{atm}(^{18}O)$ ($\pm 1\sigma$). Same average seasonal cycle of $\Delta\delta_{atm}(^{18}O)$ simulated by the box model as in (b), and corresponding $\delta_{LW}(^{18}O)$ values (blue solid lines). Additional simulations of average seasonal $\Delta\delta_{atm}(^{18}O)$ cycles and the incorporated seasonally varying $\delta_{LW}(^{18}O)$ values for sensitivity tests (thick blue dashed, two-dot chain, and dotted lines), and those incorporated the average monthly $\delta_{LW}(^{18}O)$ around TKB (36° N, 140° E)**

**(green open circles) and at lower latitude (30°S – 30°N) (red open circles) during 2013–2022 calculated by MIROC5-iso are also shown (red and green dashed lines, respectively).**

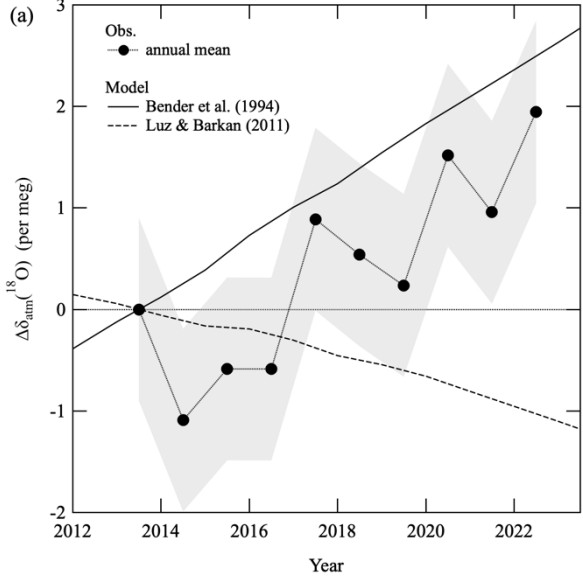

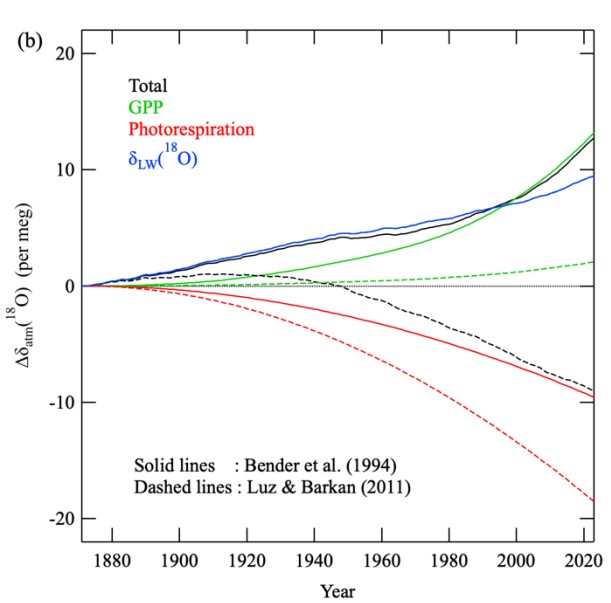

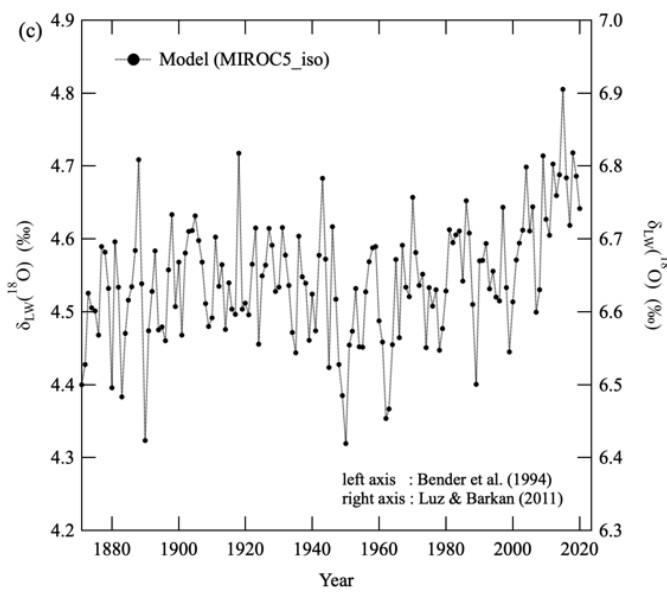

Figure 8: (a) Annual average values of $\Delta\delta_{atm}(^{18}O)$ observed at TKB (filled circles). $\Delta\delta_{atm}(^{18}O)$ simulated by the box model for the period 2012–2022, assuming the isotopic effects reported by Bender et al. (1994) (solid line) and Luz and Barkan (2011) (dashed line). $\Delta$ denotes deviations from the observed value in 2013. See text for details. (b) Same $\Delta\delta_{atm}(^{18}O)$ simulated by the box model but for data during 1871–2022 and the respective contributions of the changes of GPP (green line), photorespiration (red line), and $\delta_{LW}(^{18}O)$ (blue line) to the simulated $\delta_{atm}(^{18}O)$. Solid and dashed lines denote the simulated data assuming the isotopic effects reported by Bender et al. (1994) and Luz and Barkan (2011), respectively. The blue solid and dashed lines almost overlap. $\Delta$ denotes deviations from the simulated value in 1871. (c) Annual average values of the global average $\delta_{LW}(^{18}O)$ calculated by MIROC5-iso. The value in 1871 were arbitrarily adjusted to 4.4 (left axis) or 6.5 ‰ (right axis), respectively, for the DME in steady state by Bender et al. (1994) or Luz and Barkan (2011).

**Table A1: A list of the symbols used in the main text**

| Symbol | Definition | Unit |
|---|---|---|
| $\delta_{atm}(^{18}O)$ | $^{18}O/^{16}O$ ratio of atmospheric $O_2$ | - |
| $\delta_{LW}(^{18}O)$ | $\delta(^{18}O)$ of leaf water | - |
| $\delta_{CO2}(^{18}O)$ | $^{18}O/^{16}O$ ratio of atmospheric $CO_2$ | - |
| $\delta_{precip}(^{18}O)$ | $^{18}O/^{16}O$ ratio of precipitation | - |
| $\delta_{atm\_summer}(^{18}O)$ | $\delta_{atm}(^{18}O)$ in summer | - |
| $\delta_{atm\_winter}(^{18}O)$ | $\delta_{atm}(^{18}O)$ in winter | - |
| $\delta(O_2/N_2)$ | atmospheric $O_2/N_2$ ratio | |
| $\delta_{BIO}(O_2/N_2)$ | variations of the $\delta(O_2/N_2)$ driven by the total activities of the terrestrial and marine biosphere | - |
| $\delta_{FF}(O_2/N_2)$ | variations of $\delta(O_2/N_2)$ driven by fossil fuel combustion | - |
| $\delta(Ar/N_2)$ | atmospheric $Ar/N_2$ ratio | - |
| $y(O_2)$ | $O_2$ amount fraction | $\mu mol\ mol^{-1}$ |
| $y(O_2)_{winter}$ | atmospheric $O_2$ amount fraction in winter | $\mu mol\ mol^{-1}$ |
| $y(CO_2)$ | $CO_2$ amount fraction | $\mu mol\ mol^{-1}$ |
| $\Delta y(CO_2, B)$ | changes in the amount fraction of $CO_2$ driven by terrestrial biosphere activities | $\mu mol\ mol^{-1}$ |
| $\Delta y(CO_2, FF)$ | changes in the amount fraction of $CO_2$ driven by fossil fuel combustion | $\mu mol\ mol^{-1}$ |
| GPP | gross primary production | $Pg\ a^{-1}$ (C equivalents) |
| $P_T$ | terrestrial $O_2$ production | $Pmol\ a^{-1}$ |
| $r_{MR}$ | relative ratio for Mehler reaction | - |
| $r_{PR}$ | relative ratio for photorespiration | - |
| $r_{DR}$ | relative ratio for dark respiration | - |
| $\varepsilon_{MR}$ | Isotopic effect of Mehler reaction | - |
| $\varepsilon_{PR}$ | Isotopic effect of photorespiration | - |
| $\varepsilon_{DR}$ | Isotopic effect of dark respiration | - |
| $\varepsilon_{LE}$ | Isotopic effect of leaf water enrichment | - |
| $\varepsilon_{OR}$ | Isotopic effect of marine respiration | - |
| $\varepsilon_{TS}$ | Isotopic effect of air exchange from troposphere to stratosphere | - |

| | | |
|---|---|---|
| $\varepsilon_{ST}$ | Isotopic effect of air exchange from stratosphere to troposphere | - |
| $\varepsilon_{FF}$ | Isotopic effect in fossil fuel combustion | - |
| $R_{Res}$ | relative ratios of the annual fluxes of $O_2$ from terrestrial respiration | $a^{-1}$ |
| $R_{PS}$ | relative ratios of the annual fluxes of $O_2$ from terrestrial production | $a^{-1}$ |
| $R_{OR}$ | relative ratios of the annual fluxes of $O_2$ from marine respiration | $a^{-1}$ |
| $R_{OP}$ | relative ratios of the annual fluxes of $O_2$ from marine production | $a^{-1}$ |
| $R_{TS}$ | relative ratios of the annual fluxes of $O_2$ from troposphere to stratosphere | $a^{-1}$ |
| $R_{ST}$ | relative ratios of the annual fluxes of $O_2$ from stratosphere to troposphere | $a^{-1}$ |
| $R_{FF}$ | relative ratios of the annual $O_2$ consumption by fossil fuel combustion | $a^{-1}$ |
| $R_{LW}$ | isotopic ratio of the leaf water (note that it is not "$\delta$") | - |
| $V_L$ | volume of leaf water | $m^3$ |
| $A_L$ | area of leaf surface | $m^2$ |
| $T$ | transpiration flux | $kg\ m^{-2}\ s^{-1}$ |
| $I_{LA}$ | leaf area index | - |
| $D$ | liquid diffusivity of an isotope | $m^2\ s^{-1}$ |
| $\tau$ | crookedness of a leaf | $Kg\ m^{-1}$ |
| ER | $O_2$ and $CO_2$ exchange ratio between the atmosphere and organisms or ecosystems | - |
| OR | oxidative ratio expected from the stoichiometry of specific materials | - |
| $OR_{FF}$ or $\alpha_F$ | OR for fossil fuel combustion | - |
| $OR_B$ or $\alpha_B$ | OR for activities in the terrestrial biosphere | - |
| $\alpha_{obs}$ | observed ER | - |
| $\phi$ | ratio of photorespiration to carboxylation | - |
| $V_{O\_max}$ | maximum oxygenation velocity | $\mu mol\ m^{-2}\ s^{-1}$ |
| $V_{C\_max}$ | maximum carboxylation velocity | $\mu mol\ m^{-2}\ s^{-1}$ |
| $K_C$ | Michaelis-Menten constants for carboxylation | $\mu mol\ mol^{-1}$ |
| $K_O$ | Michaelis-Menten constants for oxygenation | $\mu mol\ mol^{-1}$ |

1000