# Peer review of "Diurnal, seasonal, and interannual variations in $\delta(^{18}O)$ of atmospheric $O_2$ and its application to evaluate natural/anthropogenic changes in oxygen, carbon, and water cycles"

_EGUsphere, 2024_

## Author Comment (AC1)

Review of "Diurnal, seasonal, and interannual variations in δ(18O) of atmospheric O2 and its application to evaluate changes in oxygen, carbon, and water cycles" by Ishidoya et al., 2024

General:

The manuscript presents a study of highest relevance for the linkages among the carbon, oxygen and water cycles. For the first time, a research group has analysed long-term measurements of the atmospheric oxygen isotope ratio ($^{18}O/^{16}O$) intending to see the impact of natural (biospheric fluxes) and anthropogenic (fossil fuel combustion). It is a fascinating manuscript to read and I would like to congratulate them for their long-term measurement effort as well as their in-depth analyses.

The manuscript is nicely written and organized and can be considered for publication with minor changes outlined below.

Thank you very much for your significant and useful comments on the paper "Diurnal, seasonal, and interannual variations in $\delta(^{18}O)$ of atmospheric O$_2$ and its application to evaluate natural/anthropogenic changes in oxygen, carbon, and water cycles" by Ishidoya et al. We have revised the manuscript, considering your comments and suggestions. Details of our revision are as follows. The line numbers denote those of the revised manuscript.

Major points:

(1) As already estimated in former studies, the expected signal in the atmospheric $\delta(^{18}O)$ by the above mentioned isotope fluxes is minimal. This requires high precision measurements over an extended period in order to see such a small signal as the single uncertainty is about 10 times larger. The required precision and accuracy, especially important for trend analysis, can only be achieved by averaging over many measurements as shown in the manuscript.

I am not sure whether the mass spectrometer was dedicated only to these measurements and whether it was run day and night autonomously. Controlling the long-term stability only every month is quite rare because the mass spectrometers behavior can change suddenly due to maintenance (filament change, ion source tuning etc). Can you please comment on these points and maybe add an additional short section about it.

Lines 99-109: Following sentences have been added considering your comments.

"In general, mass spectrometers behavior can change suddenly due to maintenance, such as filament change, ion source tuning etc.. To minimize the uncertainties associated with the changes in the

conditions of the mass spectrometer, we used the specific filaments for the measurements of air samples with the atmospheric level amount fraction of $O_2$ supplied by the Thermo Scientific co.. This enabled us to carry out the continuous measurements in the present study for 11 months without exchanging the filament (when we used the original filament supplied for the mass spectrometer, then we needed to exchange it every 3 months). After the exchange of the filament, several weeks are needed to stabilize the condition of the ion source of the mass spectrometer by flowing the sample and reference air, especially for the elemental ratios as $O_2/N_2$, $Ar/N_2$, and $CO_2/N_2$. Once the condition is stabilized, we did not tune the ion source throughout the period using the same filament. Furthermore, the mass spectrometer was dedicated only to the measurements of $\delta_{atm}(^{18}O)$ and related components including those for flask samples (e.g. Ishidoya et al., 2021, 2022) and it was run day and night autonomously to keep the condition of the ion source."

(2) Regarding the long-term estimated change in $\delta^{18}O(O_2)$, the authors assume a constant increase rate of the GPP per year. For me, this does not make sense as there are many studies out there discussing the $CO_2$ fertilization effect. Therefore, I would rather assume a scaling based on the excess $CO_2$ level ($CO_2$actual – $CO_2$ preindustrial). Even though this will most probably not affect their results significantly (Fig. 8), but it still better than use a constant increase rate. In particular as the authors have used a $CO_2$-dependent photorespiration rate (eq. 5).

Lines 461-469 and Fig. 8: We have assumed a scaling of GPP based on the $CO_2$ amount fractions from Scripps $CO_2$ Program, considering your suggestion. Specifically, following sentences have been added and the simulated $\delta_{atm}(^{18}O)$ plotted in Fig. 8 has been revised.

"We first assumed that the global terrestrial GPP increases in proportion to the global average $CO_2$ amount fraction. As the global average $CO_2$ amount fraction, we used the data from Scripps $CO_2$ Program (https://scrippsco2.ucsd.edu/data/atmospheric_co2/icecore_merged_products.html) based on ice-core data and direct observations before and after 1959, respectively (Keeling et al., 2001; Rubino et al., 2019). We assume the initial terrestrial production of $O_2$ in 1871 as 16.7 Pmol $a^{-1}$ (Table 1), which corresponds to 107 Pg $a^{-1}$ (C equivalents) of global terrestrial GPP considering the $r_{DR}$ of 0.59 and the $OR_B$ of 1.1. Then, the GPP increased secularly with increasing $CO_2$ amount fraction, and it takes 141 Pg $a^{-1}$ (C equivalents) in 2006. Although this is somewhat larger than the average GPP of 125 Pg $a^{-1}$ (C equivalents) for the period 1992–2020 reported by Bi et al. (2022), it falls within a range of the global GPP estimates from various models summarized in Fig. 10 of Zheng et al. (2020)."

Minor points:

Title:

Diurnal, seasonal, and interannual variations in $\delta(^{18}O)$ of atmospheric $O_2$ and its application to evaluate

changes in oxygen, carbon, and water cycles.

What kind of changes do you mean here? Changes in trend, seasonality, fossil fuel influence, natural changes?

Title: The title has been changed to "Diurnal, seasonal, and interannual variations in $\delta(^{18}O)$ of atmospheric $O_2$ and its application to evaluate natural/anthropogenic changes in oxygen, carbon, and water cycles".

L15-16

the amplitude is very small, how about its uncertainty? Because it is a mean of about 11 seasons, the seasonality could be smoothed.

Lines 387-390: The uncertainties for the seasonal amplitudes of $\delta_{atm}(^{18}O)$ and $\delta(O_2/N_2)$ have been added as "The minimum of the seasonal $\delta_{atm}(^{18}O)$ cycle appeared in late summer to early autumn, and the peak-to-peak amplitude was 2.1±0.6 per meg. The maximum of the seasonal $\delta(O_2/N_2)$ cycle occurred in summer, and its peak-to-peak amplitude was 113±10 per meg. The uncertainties for the amplitudes of $\delta_{atm}(^{18}O)$ ($\delta(O_2/N_2)$) was evaluated as a standard deviation of the 10-year average monthly mean values from the best-fit curve shown in Fig. 7b.". In the abstract, we did not state the uncertainty since we use the phrase "about 2 per meg".

L18

The secular increase is even more delicate to determine, which requires an extreme stability of the instrument and the standard gas measurements. What about the influence of filament changes, power interruptions, ion source tunings, inlet system, gas flow regime. I am amazed about the stability that is required. Isotope ratio may be less prone to changes but elemental ratios as $O_2/N_2$ or $Ar/N_2$ are generally more dependent on such changes. Can you comment on these. Thank you.

As our reply to your "major points (1)", we have added related sentences to lines 99-109. We note that the gas introduction methods (Lines 80-86) are primarily important to obtain the stability (details of the system can be found in Ishidoya and Murayama (2014)).

L26-28

consider rewording to:

The $^{18}O/^{16}O$ ratio of atmospheric $O_2$, $\delta_{atm}(^{18}O)$, is about 24 ‰ higher than that of ocean water (per definition 0 ‰ on the Vienna-Standard Mean Ocean Water (V-SMOW)) due to various processes in the global oxygen and water cycle (e.g. Craig, 1961; Barkan and Luz, 2005)

Lines 28-30: The sentence has been rewritten, as suggested.

L35-36

Please give corresponding references. There are many more than given here.

Lines 39-40: Some corresponding references have been added, as suggested.

L37

..of present air, ...

Line 41: The words "of air at present" have been changed to "of present air".

L37

….and that variations of the DME from the average are ±0.2 ‰. This addition is not clear, please be more specific here.

Lines 41-44: The sentences have been rewritten as follows, considering your comments.

"B94 have reported that the DME is on average lower by 0.05 ‰ than that of present air during the past 130,000 years, and the standard deviation of the DME from the average was ±0.2 ‰. They suggested that the DME was nearly unchanged between glacial maxima and interglacial periods, and the variability is small and may be due to variations of the relative rates of primary production on the land and in the ocean."

L45

Which value is now used? You may write...and have obtained a range of 22.4 to 23.3 for DME.

Lines 50-51: The words "…and have reproduced the average DME of 22.4 or 23.3 ‰" have been changed to "…and have obtained the average DME of 22.4 to 23.3 ‰d", as suggested.

L63-65

This is a very interesting statement.

Thank you very much for your evaluation.

L82-83

Switch sentence structure, 2nd part first and vice versa.

For the continuous measurements of stable isotopic ratios of $O_2$, $N_2$, and Ar ($\delta_{atm}(^{18}O)$, $\delta_{atm}(^{15}N)$, and $\delta_{atm}(^{40}Ar)$) as well as the $O_2/N_2$ ratio and amount fraction of $CO_2$, we repeatedly conducted alternate analyses of the sample and reference air.

Lines 88-89: The sentence has been modified following your suggestion as "We repeatedly conducted alternate analyses of the sample and reference air, for the continuous measurements of stable isotopic

ratios of $O_2$, $N_2$, and Ar ($\delta_{atm}(^{18}O)$, $\delta_{atm}(^{15}N)$, and $\delta_{atm}(^{40}Ar)$) as well as the $O_2/N_2$ ratio and amount fraction of $CO_2$".

L84

how come to determine a trend of 0.22 per meg or a seasonality of 2 per meg with a standard deviation of 20 per meg. This requires an well-defined long-term stability.

As described in the text (Lines 119-123), we evaluated the long-term stability from $\delta_{atm}(^{18}O)$ of three secondary standards against the primary standard air. Variations of the annual average $\delta_{atm}(^{18}O)$ of our three secondary standards were ±0.9 per meg on average (Fig. 2), which corresponds to uncertainty of ±0.13 per meg $a^{-1}$ for the 10-year-long secular trend. This enables us to detect the average trend of 0.22 per meg per meg $a^{-1}$ for the period 2013-2022. On the other hand, variations of each value of $\delta_{atm}(^{18}O)$ of our three secondary standards were ±2.3 per meg on average (Fig. 2). Therefore, it is difficult to detect a seasonality of 2 per meg of $\delta_{atm}(^{18}O)$ from 1-year observation only, but we can evaluate it as 10-year average seasonal cycle (±2.3 /sqrt(10) = ±0.7 per meg, which is consistent with the 2.1±0.6 per meg evaluated as a standard deviation of the 10-year average monthly mean values from the best-fit curve shown in Fig. 7b).

L86

…calculated by ...Keeling…

Line 92: The words "$\delta_{atm}(^{18}O)$ by Keeling…" have been changed to "$\delta_{atm}(^{18}O)$ calculated by Keeling…", as suggested.

L88-89

For this purpose, the measured values of the $\delta_{atm}(^{18}O)$ for the same air sample needed to not show any temporal drift, at least during the averaging period.

not clear what you want to say here, maybe you combine it with the previous sentence to

This averaging results theoretically in a standard error of the observed $\delta_{atm}(^{18}O)$ of less than 0.6 per meg assuming no temporal drift during the averaging period.

Lines 93-94: The sentences have been combined and rewritten as you suggested.

L106

…an uncertainty of ±0.13 per meg $a^{-1}$….how was this calculated?

Line 122: The uncertainty was calculated by $\pm\sqrt{(0.9)^2 + (0.9)^2}/10$ considering the long-term

stability (1 standard deviation) of the annual average of $\delta_{atm}(^{18}O)$ of three secondary standards (blue circles shown in Fig. 2).

L113-114

As seen in Figure 3a, $\delta_{atm}(^{18}O)$ increased linearly with increasing amount fractions of $CO_2$.

Why, what are the reasons? There is no isobaric interference. Has it to do with isotope exchange between $CO_2$ and $O_2$? Have you done $CO_2$ additions with $O_2$ labelling?

Lines 132-135: Unfortunately, the mechanism has not been clarified yet, so that we have added following sentences.

"The mechanism of the positive correlation between the $\delta_{atm}(^{18}O)$ and $CO_2$ was not clarified yet since there is no isobaric interference. In this regard, I found no significant influences of $CO_2$ amount fraction on $\delta_{atm}(^{18}O)$ for a different mass spectrometer, Finnigan MAT-252 (Ishidoya, 2003). This suggest that the influences should be examined carefully for each mass spectrometer."

We agree with you that $CO_2$ additions with $O_2$ labelling will be a useful method to evaluate the possibility of isotope exchange between $CO_2$ and $O_2$. We would like to leave it as a future task.

L120-121

reword to ….....in our earlier flask studies in 2013.

Line 141: The words "in our earlier experiments in 2013 that involved use of flasks" have been changed to "in our earlier flask studies in 2013", as suggested.

L135-136

$R_{TS}$ and $R_{ST}$ denote the ratios of the annual fluxes of $O_2$ between the troposphere and stratosphere, respectively.

to what? It is a ratio.

Lines 168-170: These are the ratios to the total amount of $O_2$ in the atmosphere. We have revised the sentence as "$R_{Res}$, $R_{PS}$, $R_{OR}$, and $R_{OP}$ represent the relative ratios of the annual amounts of $O_2$ from terrestrial respiration, marine respiration, terrestrial production, and marine production, respectively, to the total amount of $O_2$ in the atmosphere (=3.706 x $10^4$ Pmol)."

L140

…the amount fraction of $O_2$ calculated by the box model was converted to $\delta(O_2/N_2)$.

how? Assuming a norm atmosphere or using the measurements to do it correctly. I ask this because of

dilution effects.

Line 183: In the box model, the amount fraction of $O_2$ was calculated assuming a norm atmosphere.

L152-153

Here, $\varepsilon_{ST}$ was set to –4 per meg so that the diminution of $\delta_{atm}(^{18}O)$ at equilibrium was –0.4 ‰.

How come?

Lines 195-198, Table 1: In this revision, we have slightly changed the value of the stratospheric diminution, taking L&B2011 results into account. We have added Table 1 to show budgets (fluxes) and isotopic effects of atmospheric $O_2$ used in the box model. As shown in the table, stratosphere – troposphere exchange ($R_{ST}$ = 3.0 x $10^3$ Pmol a$^{-1}$) is about 100 times larger than the total biospheric flux at the surface (16.7 + 9.8 Pmol a$^{-1}$). Therefore, $\varepsilon_{ST}$ was set to 2.5 per meg to contribute to DME by about –0.3 per mil; –0.3 = $R_{ST}$ x $\varepsilon_{ST}$ / ($R_{PS}+R_{OP}$).

L161-162

This uncertainty complicates the problem of inter-annual $\delta_{atm}(^{18}O)$ change and suggests that gravitational separation may be involved in small fluctuations in the DME.

One needs to look into $O_3$ and $^{14}C$ variations at high altitudes, ideally close to the tropopause.

We agree with you associated with the importance of S-T exchange through the tropopause for isotopically light $O_2$ in the stratosphere, but in this paragraph, we discuss gravitational separation which is not necessarily reflect STE alone, but is also influenced by the balance between molecular and eddy diffusion and/or strength of Brewer-Dobson circulation (e.g. Ishidoya et al., 2021). Therefore, we leave the sentence as it is.

L204-205

…and artificial inlet fractionation induced by radiative heating of an air intake (e.g., 205 Blaine et al., 2006).

You mentioned that thermal diffusion is not affecting the measurements due to the high flow rate.

Line 268: The words "…diurnal $\delta(Ar/N_2)$ cycle, which is driven by" have been changed to "…diurnal $\delta(Ar/N_2)$ cycle, which is potentially driven by", to clarify both the night-time vertical temperature gradient and artificial inlet fractionation are just the possible causes.

L234

1.46 ….in graph 1.45

Line 299: The number was corrected to 1.45. Thank you for pointing it out.

L239

why only to terrestrial and not to marine biosphere activities?

$O_2$ and $CO_2$ fluxes between the terrestrial biosphere and the atmosphere are tightly correlated with each other (with ER ~ 1.1), while those between the ocean and the atmosphere are not due to carbonate dissociation effect. Therefore, we consider the summertime diurnal $\delta(O_2/N_2)$ with the ER of 1.08 could be attributed mainly to terrestrial biosphere activities.

L243-244

(https://www.enecho.meti.go.jp/statistics/energy_consumption/ec002/results.html#headline2,  last access: 28 March 2024, in Japanese) (Ishidoya et al., 2020).

paper in 2020, reference in 2024?

We found the past URL in Ishidoya et al. (2020) was not convenience of the readers to find specific data, so that we have shown the updated URL accessed recently on 28 March 2024.

L244-246

The implication is therefore that the isotopic discrimination of $O_2$ during activities of the terrestrial biosphere was the main cause of the observed summertime diurnal $\delta_{atm}(^{18}O)$ and $\delta(O_2/N_2)$ cycles, and the isotopic discrimination of $O_2$ during fossil fuel combustion was very small or negligible.

The same conclusion could be drawn by radiocarbon measurements. I guess $^{14}C$ measurements are being done at your station. Why not use and show it?

We agree with your suggestion, but unfortunately, our institute has not observed $\Delta^{14}C$ of $CO_2$. I guess National Institute for Environmental Studies (NIES) observed $\Delta^{14}C$ at TKB, so that the results shown in the present study will be a useful tool in future to validate $\delta_{atm}(^{18}O)$ and $\Delta^{14}C$ methods with each other.

L254-256

This method, hereafter referred to as the "$\delta_{atm}(^{18}O)$-method", enabled us to remove the impact on $\delta(O_2/N_2)$ of not only the activities of the terrestrial biosphere but also the contributions due to the air–sea $O_2$ flux, which is driven mainly by activities in the marine biosphere (e.g., Nevison et al., 2012; Eddebbar et al., 2017), from the estimated $\delta_{FF}(O_2/N_2)$.

Not clear as you first make the balance between observed and bio to obtain the FF. By doing this you

cannot disentangle the air-sea $O_2$ flux from the terrestrial $O_2$ flux.

We did not separate (disentangle) the air-sea $O_2$ flux from the terrestrial $O_2$ flux in this case. Instead, we separate the contribution of "the air-sea $O_2$ flux + the terrestrial $O_2$ flux" from that of fossil fuel combustion. This is based on (1) the simulated diurnal cycle of $\delta_{atm}(^{18}O)$ and the $\delta_{atm}(^{18}O) / \delta(O_2/N_2)$ ratio for the case considering terrestrial processes only were very similar to those for the case considering marine processes only, for both case the isotopic effects from B94 and L&B11 were incorporated into the box model (lines 312-315), (2) the contributions due to the air–sea $O_2$ flux is considered to be driven mainly by activities in the marine biosphere (e.g., Nevison et al., 2012; Eddebbar et al., 2017), and (3) the isotopic discrimination of $O_2$ during fossil fuel combustion was very small or negligible (Fig. 4). Then, we can estimate the variations of the observed $\delta(O_2/N_2)$ driven by the total activities of the terrestrial and marine biosphere ("$\delta_{BIO}(O_2/N_2)$") by dividing the observed variations in $\delta_{atm}(^{18}O)$ ($\delta_{atm}(^{18}O)$ is driven by the total activities of the terrestrial and marine biosphere) by the ratio of the simulated $\delta_{atm}(^{18}O) / \delta(O_2/N_2)$.

L267-268

It is noteworthy that propane ($CH_3CH_2CH_3$), for which the $OR_{FF}$ is 1.67 for complete combustion, should also be considered as the household gas consumed in the TKB area.

This is very interesting.

Thank you very much for your interest.

L278-281

y. Similar separation has been carried out for $CO_2$ based on the simultaneous analysis of the $\Delta(^{14}C)$ and amount fraction of $CO_2$ (e.g., Basu et al., 2020) or based on the simultaneous analysis of $\delta(O_2/N_2)$ and the amount fraction of $CO_2$ by assuming an average $OR_{FF}$ based on a statistical assessment (Pickers et al., 2022).

There are more publications available that might be cited!

Lines 362-364: We have added some references to be cited.

L291-292

Figure 7a therefore shows 116 and 120 $\delta_{atm}(^{18}O)$ and $\delta(O_2/N_2)$ data, respectively.

rewrite

Lines 374-375: The sentence has been rewritten as "Therefore, Fig. 7a shows 116 and 120 data of $\delta_{atm}(^{18}O)$ and $\delta(O_2/N_2)$, respectively."

L306-309

Keeling (1995) expected $\delta_{atm}(^{18}O)$ to be lower in summer than in winter by 2 per meg based on the assumption that the 100 per meg seasonal increase of $\delta(O_2/N_2)$ was driven by the input of photosynthetic $O_2$, the $\delta(^{18}O)$ of which is about 20 ‰ lower than $\delta_{atm}(^{18}O)$.

Show how to calculate it!
Lines 393-398: The calculation method has been added, as suggested.

L316

We found that the box model could reproduce the observed seasonal $\delta_{atm}(^{18}O)$ cycles

You adjusted the corresponding values. Questions are remaining as to whether the used model values fall within known ranges.
Table 1: We have added Table 1 to clarify the specific values and references for the parameters incorporated into the box model, considering your comments.

L355-357
see major point 2

Fig. 2
The measurements are not equally distributed over time, this influences the uncertainty per year. Have you considered this?
We did not consider an effect of the non-uniform distribution you pointed out. I agree this could influence the uncertainty for secular trend, nevertheless the effect is not so serious since the measurements in 2012-2014 (the first period) and those after 2020 (the last period) is relatively denser than those in 2014-2020 as seen from Fig. 2. The uncertainty of the average secular trend throughout the period is determined mainly by the measurements in the first and last periods.

---

## Author Comment (AC2)

The authors present measurements of the $^{18}O$ isotopic composition of atmospheric molecular oxygen from air taken at the roof of their building in the greater Tokyo region. They also measured $O_2/N_2$ ratios and $CO_2$ concentrations to assess the suitability of delta-$^{18}O$ in $O_2$ to probe the carbon cycle. They analyse the diurnal and seasonal variations as well at the interannual trend with a one-box model. The observations are pretty exciting. I am not aware of any group that measured the diurnal cycle of delta-$^{18}O$ in atmospheric $O_2$ before. Given the tremendous technical advances over the last years, it seems logical that they succeeded, eventually. They still have to average about 1000 individual measurements so that average diurnal cycles for each season are presented.

I am less convinced by the analysis with the one-box model. The method was not well explained, so that it is possible that I missed some things. I think that the manuscript needs some serious revisions before publication.

Thank you very much for your significant and useful comments on the paper "Diurnal, seasonal, and interannual variations in $\delta(^{18}O)$ of atmospheric $O_2$ and its application to evaluate natural/anthropogenic changes in oxygen, carbon, and water cycles" by Ishidoya et al. We have revised the manuscript, considering your comments and suggestions. Details of our revision are as follows. The line numbers denote those of the revised manuscript.

Why are there only the values of Bender et al. (1994) but not the updated figures of Luz and Barkan (2011, doi: 10.1029/2010GB003883)?

Section 2.2 (lines 143-236) and Table 1: Considering your comment, we have carried out the simulations by using the isotope effects not only from Bender et al. (1994) but also from Luz and Barkan (2011) (referred to as "B94" and "L&B11", respectively, in the manuscript), and compared them especially for the long-term variations of $\delta_{atm}(^{18}O)$. Table 1 has also been added to summarize values of $O_2$ budgets and isotopic effects for both studies.

Where is fossil fuel in the box model? It looks like the model comes from Bender et al. (1994) who analysed the last 130,000 years and did not need fossil fuel.

Equations (3) and (4) and related sentences, lines 172-178, and lines 322-329: In the revised manuscript, contributions of fossil fuel combustion has been included in the equations for the box model. However, we assume that atmospheric oxygen is consumed without isotope effects in fossil fuel combustion considering high temperature during the industrial combustion processes. We have added some sentences to note future needs to examine the combustion processes as follows.

"(lines 172-178) We assumed that atmospheric oxygen is consumed without isotope effects in fossil fuel combustion ($\varepsilon_{FF}=0$), taking into account that the industrial combustion processes usually occur at high temperature. Therefore, we consider no contribution to DME from fossil fuel combustion in this

study. In this regard, it is known that large oxygen isotope fractionation occurs in the combustion processes such as biomass burning due to complex combustion processes (Schumacher et al., 2011). In such cases, it will be necessary to consider isotopic fractionation in the consumption of atmospheric oxygen associated with combustion. However, at present, little is known about the impact of this on DME."

"(lines 322-329) It would be generally reasonable since the combustion occurs at high temperature, which minimizes isotopic discriminations. However, Schumacher et al. (2011) reported isotopic discriminations on the order of up to 26 ‰ for stable oxygen isotopic ratio of atmospheric $CO_2$ ($\delta_{CO2}(^{18}O)$) derived from combustion of different kinds of material. They suggested that natural combustion processes on the long term might enrich $\delta_{atm}(^{18}O)$ and contribute to the DME. Therefore, isotopic discriminations of $\delta_{atm}(^{18}O)$ due to combustion processes should be examined carefully in future, based on precise observations of $\delta_{atm}(^{18}O)$."

The method of ignoring terrestrial or marine fluxes was not well explained. Only the terrestrial fluxes had a sinusoidal cycle in the methods. So how does the model calculate a diurnal cycle if terrestrial $O_2$ fluxes are ignored?

Lines 231-233: The sentence "We also carried out simulations of diurnal changes considering marine $O_2$ consumption and production approximated by the similar simple function, to examine sensitivities of $\delta_{atm}(^{18}O)$ / $\delta(O_2/N_2)$ ratio to the terrestrial and marine signals" has been added to make the method clearer.

The rationale behind their calculations of the delta-$^{18}O\_atm$-method were not given. Why does the ratio give you delta-$^{18}O\_bio$? Is it because fossil fuel is missing in the box model?

As you expected, it is because we ignored the isotopic effect of fossil fuel combustion. Regarding this assumption, we have added some sentences to note future needs to examine the processes (please check our reply to your comments "Where is fossil fuel in the box model?..." above).

The box model has a shifted diurnal cycle by about two hours. You would get a strange signal when dividing two sinusoidal signals shifted by some delta. This does not seem the case here and I was wondering why? Is it possible that you get a false diurnal cycle of the biospheric fluxes because of the wrong timing of the box model?

In the simulations, diurnal cycles of respiration and photosynthesis fluxes were approximated by simple sinusoidal signals. We could not know true phases and amplitudes of the diurnal cycles, so that we adjusted the amplitude of the respiration and photosynthesis fluxes arbitrarily under the constraint that both fluxes became the largest around noon. Therefore, if we shift the peaks of the fluxes by two hours earlier, then we can reduce the phase shifts of the simulated $\delta_{atm}(^{18}O)$ and $\delta(O_2/N_2)$ found in Fig.

4a. However, we consider it would be reasonable that the largest respiration and photosynthesis fluxes are found around noon, so that we leave the simulated diurnal cycles in Fig. 4a as they are.

The ER method is not explained. Where are the numbers 1.1, 1.4-1.7 coming from? I guess nowadays 1.05 is more accepted for photosynthesis.

Lines 335-338: The sentence has been modified considering your comments as "…we assumed the OR for activities in the terrestrial biosphere ($OR_B$) to be 1.1 (Severinghaus, 1995), which has been widely used in past studies (e.g. Manning and Keeling, 2006; Tohjima et al.), and the $OR_{FF}$ to be 1.4, 1.5, 1.6, or 1.7 considering mixed combustion of solid fuel, liquid fuel, and natural gas. It is noted some recent studies have used the $OR_B$ of 1.05 rather than 1.1 (e.g. Morgan et al., 2021)".

How is it possible that in $O_2$ leaf water isotopes are increasing by 9 per meg (Figure 8) when leaf water isotopes of $H_2O$ are increasing by nothing until 2000 and then only about 0.2 permil?

Lines 512-516: Following sentences have been added to show the reason clearer.

"It is noted that the contributions of the changes of $\delta_{LW}(^{18}O)$ to the simulated $\delta_{atm}(^{18}O)$ increased with time monotonously while clear increase of $\delta_{LW}(^{18}O)$ was found after the 1980s (Figs. 8b–c). This is due to the choice of the initial $\delta_{LW}(^{18}O)$ in 1871; we set it to be 4.4 or 6.5 ‰ (the values for steady state by B94 or L&B11). As seen from Fig. 8c, the average $\delta_{LW}(^{18}O)$ during 1872–1980 was higher than the initial values, which made the monotonous increase of the $\delta_{atm}(^{18}O)$ driven by the $\delta_{LW}(^{18}O)$ changes..

What are the leaf water scenarios in Figure 7? I could not find any explanations. I would be curious how you get a time shift of up to two months from different formulations of leaf water.

Lines 421-435: The sentences have been rewritten to explain the scenarios. The scenario for the thick dashed blue line in Fig. 7 was determined based on some past studies reported seasonal variations of $\delta_{LW}(^{18}O)$, and other two scenarios represented by two-dot chain and dotted lines were carried out as sensitivity tests to the phase difference in seasonal $\delta_{LW}(^{18}O)$ cycle.

You get an increasing or decreasing secular trend if the right-hand side of Eq. (3) is non-zero. This can have many reasons and you do not have to have increasing GPP or anything. The authors have tweaked so many fluxes in their model that I do not think that we can say anything about the secular trend.

Section 3.3 (lines 441-558) and Fig. 8: We have revised the analysis and discussion for the secular trend including the simulations by the box model incorporated not only B94 but also L&B11. As you expected, the secular trends of the simulated $\delta_{atm}(^{18}O)$ are highly sensitive to the isotopic effects associated with the DME. Therefore, further studies are needed to determine the isotopic effects precisely, in order to evaluate long-term changes in GPP and photorespiration based on $\delta_{atm}(^{18}O)$.

This reminds me of the literature of delta-$^{18}$O in atmospheric $CO_2$. There were the same issues: a secular trend due to unbalanced fluxes, a one- to two-month time shift in the seasonal cycle, etc. The authors could learn at lot from that literature but not a single paper is referenced in the manuscript.

Lines 408-413 and 498-502: Following sentences have been added to note the characteristics of seasonal and interannual variations in stable oxygen isotopic ratio of atmospheric $CO_2$ ($\delta_{CO2}(^{18}O)$) for a comparison with those of $\delta_{atm}(^{18}O)$.

"(lines 408-413) In this context, seasonal cycles of $\delta_{CO2}(^{18}O)$ have been reported by some past studies (e.g. Peylin et al., 1999; Cunz et al., 2003; Murayama et al., 2010). Peylin et al. (1999) and Cunz et al. (2003) used 3-D atmospheric transport models to reproduce the observations, and they found the main contributors are respiration and production for the respective seasonal cycles of $\delta_{CO2}(^{18}O)$ and $CO_2$ amount fraction. These characteristics are different from the seasonal cycles of $\delta_{atm}(^{18}O)$ and $\delta(O_2/N_2)$ observed in this study, both of which are driven mainly by production (Fig. 7c)."

"(lines 498-502) It should be noted that Welp et al. (2011) suggested that $\delta_{CO2}(^{18}O)$ increases with increasing $\delta_{precip}(^{18}O)$ and $\delta_{LW}(^{18}O)$ through the redistribution of moisture and rainfall in the tropics during an El Niño, which leads to substantial interannual variations in $\delta_{CO2}(^{18}O)$ during 1977–2009 obtained from the Scripps Institution of Oceanography global flask network. Therefore, it will be important in future studies to examine not only secular trend discussed in this study but also interannual variations in $\delta_{LW}(^{18}O)$ and $\delta_{atm}(^{18}O)$."

Nobody thinks that GPP increase over the last century comes solely from a decrease of photorespiration. The discussion from page 14 line 397 up to page 15 line 417 is weird regarding the carbon cycle and anything we know about photosynthesis.

The discussion you pointed out in the previous manuscript has been removed from the revised manuscript.

The leaf water from MIROC5-iso looks like source water. Most global models of water isotopes do not calculate leaf water enrichment because they assume steady state so that the transpired water is the same as source water. If not, I would have loved to know how delta-$^{18}$O of leaf water is calculated in MIROC5-iso.

Lines 247-254: MIROC5-iso does calculate the isotopic fractionation effect when transpiration occurs. We added the equation to make the point clearer.

More minor comments are:

Why is the $^{18}$O in parenthesis in $\delta(^{18}O)$? This is a weird notation.

We understand your comment, however, I have used the phrase following the Editor's instruction.

Given the current notation. It is never clear which molecule is looked at. Sometimes $\delta_{LW}(^{18}O)$ is $O_2$ and sometimes $H_2O$. Perhaps making it clearer, e.g. adding the molecule behind such as $\delta^{18}O\text{-}O_2$ or $\delta^{18}O(O_2)$?

In the revised manuscript, we have used $\delta_{LW}(^{18}O)$ for $H_2O$ only. The words "…and the respective contributions of GPP (solid green line), photorespiration (solid red line), and $\delta_{LW}(^{18}O)$ (solid blue line) to the simulated $\delta_{atm}(^{18}O)$…" in caption of Fig. 8 has been changed to "…and the respective contributions of the changes of GPP (green line), photorespiration (red line), and $\delta_{LW}(^{18}O)$ (blue line) to the simulated $\delta_{atm}(^{18}O)$." to make the meaning clearer.

Lots of references are missing like all the references for the emission ratio method. Or ER = 1.67 for propane?

References for the DME, emission ratio methods, and $\delta_{CO2}(^{18}O)$ have been added in the revised manuscript (e.g. lines 39-40, 154, 176, 336-338, 362-364). As for the $OF_{FF}$ for propane, the words "…the $OR_{FF}$ is 1.67 for complete combustion" have been changed to "…the $OR_{FF}$ is 1.67 assuming complete combustion" (line 350). If propane completely burned, $C_3H_8 + 5O_2 \rightarrow 3CO_2 + 4H_2O$, then the OR is 5/3 = 1.67.

I was wondering if MIROC5-iso has no carbon cycle? Most models have nowadays. So why not using these fluxes, or at least its dirunal and seasonal variations, instead of simple sinusoidal fluxes?

Thank you for the valuable suggestion. Unfortunately, the carbon cycle is not included in MIROC5-iso, although it is included in another version of MIROC; MIROC-ESM (Watanabe et al., 2011). We would like to leave it as a future task.

---

## Author Comment (AC3)

Jeff Severinghaus,

Ishidoya et al. have produced a stunning and groundbreaking extension of the well-known millennial-scale variations in atmospheric oxygen isotopes (namely $^{18}O$ of $O_2$), sometimes known as the Morita-Dole Effect, that are recorded in ice cores. Their extension brings to the table totally new and fascinating information - namely the first high-quality observations of diurnal and seasonal cycles in $^{18}O$ of $O_2$. Their tour-de-force treatment of extremely difficult analytical techniques makes it possible now to ask totally new questions about the role of the terrestrial biosphere in the last 5 decades of (unplanned) anthropogenic $CO_2$ fertilization due to fossil fuel burning, as just one example among many.

The quality of their measurements is superb, and unparalleled. Their deep attention to details, and exploration of potential pitfalls, makes their conclusions robust and convincing.

One very minor comment I would make is that their box model estimate of ~1500 years for the turnover time of atmospheric $O_2$ may be a little too long. My ice core work shows that $^{18}O$ of atmospheric $O_2$ relaxes with a characteristic asymptotic decay curve after abrupt climate change events on a timescale of about ~1000 years, implying that the turnover time of $O_2$ in the atmosphere is about ~1000 years.

The authors are to be congratulated for a true breakthrough that will no doubt open many doors for future study of the interlinked carbon, oxygen, and argon cycles in Earth's atmosphere. These studies will no doubt shed light on the ongoing anthropogenic perturbations to the cycles of these gases.

Thank you very much for your significant and useful comments on the paper "Diurnal, seasonal, and interannual variations in $\delta(^{18}O)$ of atmospheric $O_2$ and its application to evaluate natural/anthropogenic changes in oxygen, carbon, and water cycles" by Ishidoya et al. We are so happy to hear that you are pleased with the work that was done. We have revised the manuscript, considering your comments and suggestions, and those from other two reviewers. We have added following sentences, considering your comments regarding the turnover time.

"(lines 214-221) The biospheric turnover time of $O_2$ in the steady state was 1398 years, which is longer than the 1200 years estimated by B94. This may be a little too long, since the $\delta_{atm}(^{18}O)$ variations reported by Severinghaus et al. (2009) based on the ice core measurements that showed a characteristic asymptotic decay curve after abrupt climate change events on a timescale of about ~1000 years, implying that the turnover time of $O_2$ in the atmosphere is about 1000 years. The biospheric turnover time is inversely proportional to the sum of the terrestrial and oceanic productions of $O_2$ incorporated

into the box model, which is 26.5 (16.7 + 9.8) Pmol a$^{-1}$ in this study (Table 1). This implies that total production of $O_2$ for the initial value in our model is underestimated. In this regard, turnover time decreases to about 1000 years when we simulate a case in which the GPP is increased, as will be discussed later."

---

## Author Response (AR2)

**Responses to Referee 2**

It was very good for the manuscript that the study of Luz and Barkan (2011) was included, eventually. It makes the analysis much more balanced and emphasizes that delta-18O of O2 adds another interesting piece to the puzzle of constraining carbon gross fluxes but it is not the silver bullet solution as presented before.

Thank you very much for your significant and useful comments on the paper "Diurnal, seasonal, and interannual variations in $\delta(^{18}O)$ of atmospheric $O_2$ and its application to evaluate natural/anthropogenic changes in oxygen, carbon, and water cycles" by Ishidoya et al. We have revised the manuscript, considering your comments and suggestions. Details of our revision are as follows. The line numbers denote those of the revised manuscript.

My biggest concern with the paper now are the leaf water isotopes.

The current manuscript shows the equation for canopy intercepted water, i.e. Eq. (6) of Yoshimura at al. (2006). This is not leaf water. The latter is in their Eq. (18) and some text above that references to Eqs. (5) and (6). alpha_k should, for example, not be from Merlivat and Jouzel (1979) for leaves.
As mentioned earlier, the current leaf water isotopic composition looks more like precipitation and the authors arbitrarily shifted the calculated leaf water isotopes. Is it possible that the authors just used the wrong variable from the MATSIRO-iso output? Please check, perhaps together with Kei Yoshimura.

Lines 245-255: Thank you for your comments. We double-checked and are sure that the variable used in the manuscript is correct (isotopic ratio of leaf water). Equation (5) in our previous manuscript was not an equation either for canopy water or leaf water. It was the isotope ratio of transpired water and has a similar form as Eq. (6) of Yoshimura et al. (2006) as written in Sect. 2.5 in their paper; "Transpiration flux Et estimates include the isoflux that considers equilibrium and kinetic fractionation from liquid to gas at stoma by the similar equations as (5), (6),…". With the equation, we aimed to show that the isotope ratio of transpired water is not the same as that of source water. However, this could be the cause of the confusion. For clarity, we have changed Eq. (5) from that for the isotopic ratio of transpired water to that for the isotopic ratio of leaf water.

My second point is the discussion about the seasonal cycle of leaf water isotopes. It is clear that if I change the isotopic composition of leaf water, I change the isotopic cycle without the elemental cycle. So this obvious fact is overstated in the manuscript.
It was rather interesting to see that the seasonal cycle of delta-18O did not change by more than 3 months when the cycle of leaf water isotopes was shifted throughout the whole year. So my conclusion would be that delta-18O will probably not be very suitable to constrain leaf water isotopes.

The discussion would have been better if seasonal cycles from MATSIRO-iso would have been used. The cited papers are mostly in the northern hemisphere while most photosynthesis happens in the tropics. How are leaf water isotopes there? What's the timing of the cycle?

Lines 440-450, and Fig. 7d: We have rewritten the sentences as follow considering your suggestion. The simulated seasonal $\delta_{atm}(^{18}O)$ cycles based on the $\delta_{LW}(^{18}O)$ calculated by MIROC5-iso have also been included in Figure 7d.

"We also carried out additional box-model simulations that incorporated the average monthly $\delta_{LW}(^{18}O)$ around TKB (36° N, 140° E) and at lower latitude (30°S – 30°N) calculated by MIROC5-iso for the period 2013–2022. The results are plotted in Fig. 7d, and both the monthly $\delta_{LW}(^{18}O)$ and simulated seasonal $\delta_{atm}(^{18}O)$ cycle fall within the range of those represented by blue solid, dashed, two-dot chain, and dotted lines in the figure discussed above. Another factor to change the simulated seasonal $\delta_{atm}(^{18}O)$ cycle is the choice of the isotopic effects from B94 or L&B11 (Table 1). If we use the isotopic effects from L&B11 and ignore $R_{Res}$ and $R_{PS}$ (i.e. we consider marine respiration/production only), then the seasonal amplitude of the simulated $\delta_{atm}(^{18}O)$ increase by 20% compared with that simulated by using the isotopic effects from B94. This is due to the difference in the isotopic effects of ocean respiration, which are 18.9 and 23.5 ‰ for B94 and L&B11, respectively. Such difference will become apparent in the southern hemisphere, where seasonal $\delta(O_2/N_2)$ cycle is driven mainly by air-sea $O_2$ flux (e.g. Keeling and Manning, 2014). Therefore, spatiotemporal variations in the seasonal $\delta_{atm}(^{18}O)$ cycle will be useful to constrain not only spatiotemporal variations of $\delta_{LW}(^{18}O)$ but also the isotopic effects of ocean respiration."

More minor comments are:

- It is Cuntz and not Cunz in the references.

Lines 411, 412, and 419: The word "Cunz" has been corrected to "Cuntz". Sorry for the typo.

- I found the notation R_Res, etc. not very intuitive. It is only defined by words: "annual fluxes of O2 from terrestrial respiration [...] to the total amount of O2 in the atmosphere". There is no unit given. Could there be simply fluxes (as in Table 1) and the amount goes to the left-hand-side?

Lines 169-173: The sentences have been rewritten as follows to make the meanings and units of the "$R_{xx}$" clearer.

"$R_{Res}$, $R_{PS}$, $R_{OR}$, and $R_{OP}$ (the unit is $a^{-1}$) represent the relative ratios of the annual fluxes of $O_2$ from terrestrial respiration, terrestrial production, marine respiration, and marine production, respectively, to the total amount of $O_2$ in the atmosphere (=3.706 x $10^4$ Pmol). For example, if we assume that the terrestrial flux is 16.7 Pmol $a^{-1}$, $R_{PS}$ will be 16.7/(3.706 x $10^4$) = 4.5 x $10^{-4}$ $a^{-1}$, as shown in Table 1. $R_{TS}$ and $R_{ST}$ denote the relative ratios of the annual fluxes of $O_2$ between the troposphere and

stratosphere, respectively." In addition, the definitions and units of all variables, including $R_{xx}$, are summarized in Table A1.

- Eq. (4) is odd. There is a d missing, probably. And it shows the strangeness of the R_x notation because the right-hand-side suddenly has y(O2) (after multiplying with it) while most fluxes do not depend on it.

Line 182: We have revised eq. (4) since there was a "d" missing, as you pointed out. Other terms in Eq. (4) are correct. Derivation of eq. (4) is as follow.

$$\frac{dM_{O2}}{dt} = (r_{MR} + r_{PR} + r_{DR})F_{Res} + F_{PS} + F_{OR} + F_{OP} + F_{TS} + F_{ST} + F_{FF}$$

Here, $M_{O2}$ is the total amounts of atmospheric $O_2$ (=3.706 x $10^4$ Pmol), and $F_{xx}$ are $O_2$ fluxes (Pmol a$^-$ $^1$). The ratio $F_{xx}/M_{O2}$ is $R_{xx}$.

$$\frac{1}{M_{O2}}\frac{dM_{O2}}{dt} = (r_{MR} + r_{PR} + r_{DR})R_{Res} + R_{PS} + R_{OR} + R_{OP} + R_{TS} + R_{ST} + R_{FF}$$

In the one box model, the ratio of the total amounts of $O_2$ and atmospheric molecules is $O_2$ amount fraction, so we get:

$$y(O_2) = \frac{M_{O2}}{M}$$

Here, $M$ is the total amounts of atmospheric molecules (Pmol). Because $M$ is a constant value, the time derivative is given by:

$$\frac{dy(O_2)}{dt} = \frac{1}{M}\frac{dM_{O2}}{dt} \quad ,$$

and

$$\frac{1}{M_{O2}}\frac{dM_{O2}}{dt} = \frac{1}{y(O_2)}\frac{dy(O_2)}{dt} \quad .$$

As a result, we get (Eq. 4) as follows:

$$\frac{1}{y(O_2)}\frac{dy(O_2)}{dt} = (r_{MR} + r_{PR} + r_{DR})R_{Res} + R_{PS} + R_{OR} + R_{OP} + R_{TS} + R_{ST} + R_{FF} \quad .$$

**Responses to the Editor**

Thank you very much for your significant and useful comments on the paper "Diurnal, seasonal, and interannual variations in $\delta(^{18}O)$ of atmospheric $O_2$ and its application to evaluate natural/anthropogenic changes in oxygen, carbon, and water cycles" by Ishidoya et al. We have revised the manuscript, considering your comments and suggestions. Details of our revision are as follows. The line numbers denote those of the revised manuscript.

One of the reviewers has raised some concerns about the leaf water isotopic composition. Could you please address them?
Lines 245-255, 440-450, and Fig. 7d: We have addressed the concerns. Please confirm our responses for Referee 2.

Regarding the reviewer's minor technical comments, I believe Eq. 4 is correct, but please double-check their query.
Line 182: We have revised Eq. (4) since there was a "d" missing. Other terms in Eq. (4) are correct. We have also showed derivation of Eq. (4) in our responses for Referee 2.

Regarding the symbols, there are indeed a lot of symbols used in your manuscript. Could you please include an appendix with a list of symbols used, with their meanings and definitions, and SI units?
Lines 146-147, 606-608, Table A1: Table A1 including the symbols used, with their definitions and units have been added, considering your suggestion.

I'd also like to ask for a few other technical corrections:

1) Symbols in equations should consist of a single Latin or Greek character. While the symbols OR and ER are fine as abbreviations, they should not be used in equations. Please find a suitable replacement, e.g., alpha_F and alpha_B instead of OR_FF and OR_B; alpha_obs instead of ER_obs.
Lines 342-345: We have replaced $OR_{FF}$ and $OR_B$ by $\alpha_F$ and $\alpha_B$, respectively.

2) The definition of ER and OR should include a physical quantity symbol for the amount fraction, i.e., alpha = -Δy(O2) / Δy(CO2). Chemical symbols such as O2 and CO2 should not be used to designate physical quantities.
Lines 299-301: The word "$\Delta O_2 \Delta CO_2^{-1}$" has been corrected to "$\Delta y(O_2)\Delta y(CO_2)^{-1}$".

3) For clarity and ease of reference, please be consistent in your choice of labels. For example, in Eq.

6 you use both "B" and "TB" for terrestrial biosphere, associated with different quantities. Please use the same index label.

Lines 341-347: The word "TB" has been changed to "B".

4) In Eq. 9, the symbols p(O2) and p(CO2) should be used instead of "O" and "C" as quantity symbols. "O" and "C" as index labels for the other quantities are fine.

Lines 485, 489-490: The sentences have been rewritten as "$y(O_2)$ and $y(CO_2)$ are amount factions of $O_2$ and $CO_2$, respectively, in equilibrium with their dissolved amount fractions in the chloroplast stroma. We used atmospheric $y(O_2)$, and atmospheric $y(CO_2)$ multiplied by 0.7 following B94.", and the symbols "O" and "C" in Eq. (9) have been changed to "$y(O_2)$" and "$y(CO_2)$", respectively.

5) Please explain the extraneous factor 10^(-3) in Eq. 9.

Lines 492-493: The sentence "$10^{-3}$ is a coefficient to compare the calculated $\phi$ with those in Farquhar et al. (1980) directly since they used units of mbar and μbar for the $y(O_2)$ and $y(CO_2)$, respectively" has been added.

6) Please remember to make the data available in a public repository that does not require user registration.

We have confirmed the WDCGG will allow us to deposit the monthly mean $\delta^{18}O_{atm}$, $\delta(O_2/N_2)$, and $CO_2$ amount fractions in their public repository. I will let you know the once the data become available.

---

## Author Response (AR3)

**Responses to the Editor**

Thank you very much for your significant and useful comments on the paper "Diurnal, seasonal, and interannual variations in $\delta(^{18}O)$ of atmospheric $O_2$ and its application to evaluate natural/anthropogenic changes in oxygen, carbon, and water cycles" by Ishidoya et al. We have revised the manuscript, considering your comments and suggestions. Details of our revision are as follows. The line numbers denote those of the revised manuscript.

I am pleased to accept the manuscript subject to an outstanding technical correction, i.e., the deposition of the full data set with the WDCGG data centre and provision of a DOI (digital object identifier). In addition to monthly averages, please include the full resolution dataset prior to any averaging. I understand you obtain a measurement every 62 s, so I'd expect to see data points at roughly 1 minute intervals. The full dataset is required for reproducing the derived results shown in the figures.

We have deposited full dataset in Zenodo which contains not only averaged data but also the raw data before averaging, following your comments. Therefore, we have revised the sentence of "Data availability" as "The observational data are available through Zenodo at https://zenodo.org/records/14221768". We have also checked the averaged data carefully and revised the figures very slightly in the revised manuscript. Nothing has changed in the conclusions.